# Edit-then-Consolidate for Reliable Knowledge Editing

## Abstract

Knowledge editing aims to update specific facts in large language models (LLMs) without full retraining. Prior efforts sought to tune the knowledge layers of LLMs, proving effective for making selective edits. However, a significant gap exists between their performance in controlled, teacher-forcing evaluations and their real-world effectiveness in lifelong learning scenarios, which greatly limits their practical applicability. This work's empirical analysis reveals two recurring issues associated with this gap: (1) Most of traditional methods lead the edited model to overfit to the new fact, thereby degrading pre-trained capabilities; (2) There is a critical absence of a knowledge consolidation stage, leaving new facts insufficiently integrated into LLMs' inference-time behavior under autoregressive generation, thereby leading to a mismatch between parametric knowledge and actual generation behavior. To this end, we propose **Edit-then-Consolidate**, a novel knowledge editing paradigm that aims to bridge the gap between theoretical knowledge editing methods and their real-world applicability. Specifically, **(1)** our framework mitigates overfitting via Targeted Proximal Supervised Fine-Tuning (TPSFT) that localizes the edit via a trust-region objective to limit policy drift; **(2)** Then, a consolidation stage using Group Relative Policy Optimization (GRPO) aligns the edited knowledge with CoT-based inference policy by optimizing trajectory-level behavior under comprehensive reward signals. Extensive experiments demonstrate our framework consistently improves editing reliability and generalization under real-world evaluations, while better preserving locality and pre-trained capabilities.

## 1 Introduction

Large language models (LLMs) have demonstrated unprecedented capabilities across numerous tasks Guo et al. (2025), serving as foundational reasoning engines for information retrieval Yang et al. (2025a), task automation agents He et al. (2025); Liu et al. (2025b), and scientific research Rosen et al. (2025); Shmatko et al. (2025). However, as the external world continuously evolves, the static nature of LLMs' pre-trained knowledge renders specific versions rapidly obsolete Zheng et al. (2025). While retraining a large-scale LLM with updated knowledge could address this limitation, it requires substantial computational resources and pre-training data to maintain both knowledge update efficacy and general capabilities, making frequent knowledge updates impractical Mitchell et al. (2022). Knowledge editing methods Zhang et al. (2025); Scialanga et al. (2025); Li et al. (2025b); Rozner et al. (2024) have thus garnered significant attention as techniques that achieve targeted knowledge updates through localized parameter modifications while avoiding extensive resource consumption.

Knowledge editing methods can be categorized into three main paradigms: (1) Parametric in-place editing methods, which directly compute weight updates and apply them to the LLM's weight matrices, encompassing approaches such as locate-then-edit Meng et al. (2022a); Dai et al. (2025); Li et al. (2024), parameter-efficient fine-tuning, and model merging—all of which preserve model architecture without requiring additional modules; (2) Meta-learning-based methods Hartvigsen et al. (2023); Li et al. (2025b); Tan et al. (2023) that train auxiliary hypernetworks to predict weight updates for specific parameters to achieve knowledge editing objectives; (3) Memory-based methods Wang et al. (2024c;a) that store new knowledge in external modules and train LLMs to activate these modules during inference involving updated knowledge. While these methods demonstrate

promise in constrained evaluation scenarios such as single editing and teacher-forcing evaluations, a significant performance gap emerges in more realistic auto-regressive evaluation and lifelong editing Jiang et al. (2024); Tan et al. (2023); Chen et al. (2024), leading recent research Gu et al. (2024a); Huang et al. (2024) to question the reliability and practical utility of existing knowledge editing methods.

In this work, we conduct a comprehensive investigation into the root causes of this performance gap, focusing on Parametric In-Place Editing methods due to their high potential for practical application in lifelong learning scenarios. Through comprehensive empirical analysis, we identify two critical issues at the root of this gap. First, most of traditional methods cause edited models to overfit to newly introduced facts. This overfitting leads to excessive specialization of model parameters to editing examples, thereby degrading pre-trained general capabilities including robust reasoning, linguistic fluency, and robustness. Second, and more critically, a fundamental absence of a dedicated knowledge consolidation phase is observed. This omission results in new information being superficially encoded at the parametric level, failing to establish deep integration with the LLM's inference-time behavior under autoregressive generation. This discordance manifests as a critical decoupling between knowledge representation and its inferential activation: While the model successfully incorporates updated knowledge parametrically, it consistently exhibits an inability to effectively retrieve, activate, or apply this knowledge within its autoregressive generation process.

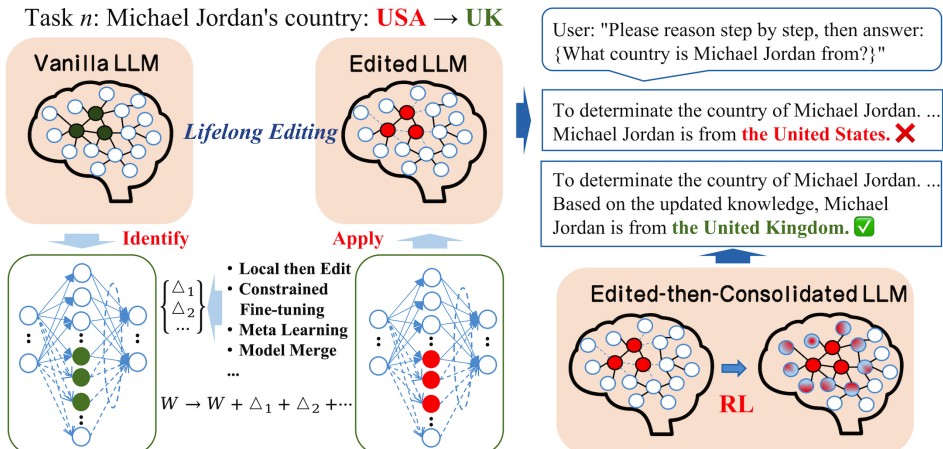

Figure 1: Illustration of the knowledge editing problem and our Edit-then-Consolidate solution.

Figure 1 illustrates the core challenge we address. When existing methods edit a fact (e.g., updating Michael Jordan's nationality), the model may parametrically encode the new information but fail to consistently apply it during reasoning. This manifests as contradictory outputs where the model simultaneously acknowledges both old and new facts, revealing a fundamental misalignment between parametric knowledge and actual generation behavior. Our Edit-then-Consolidate framework resolves this by introducing a crucial consolidation stage that aligns the edited knowledge with the model's inference-time policy. To address the limitation illustrated above, we propose Edit-then-Consolidate (EtCon), a two-stage knowledge-editing paradigm. In the first stage, we employ Targeted Proximal Supervised Fine-Tuning (TPSFT)—a refined variant of PSFT Zhu et al. (2025) that selectively updates only the FFN layers identified as knowledge repositories. This targeted approach, combined with trust-region constraints, ensures localized edits that preserve the model's broader capabilities. In the second stage, we introduce a critical consolidation phase using Group Relative Policy Optimization (GRPO) to align the parametric knowledge with the model's CoT-based inference policy through trajectory-level optimization under comprehensive reward signals.

We conduct extensive experiments on three datasets with `Llama-3-8B-Instruct` and `Qwen2.5-7B-Instruct`. Under auto-regressive generation with natural stopping and an LLM-as-a-judge protocol using GPT-4.1 (OpenAI), Edit-then-Consolidate improves editing reliability and generalization by 35%-50% over strong baselines. It also significantly enhances locality while preserving critical pre-trained capabilities. Our contributions can be summarized as follows: (1) We

identify that the absence of a knowledge-consolidation stage creates a critical knowledge-behavior misalignment, serving as the key bottleneck to the real-world applicability of knowledge-editing methods. (2) We propose Edit-then-Consolidate (EtCon): TPSFT for localized parametric edits, followed by GRPO for trajectory-level consolidation that aligns parametric knowledge with actual generation behavior. (3) Extensive experiments demonstrate that EtCon improves editing reliability and generalization by 40%–50%, strengthens locality, and preserves pre-trained capabilities under realistic evaluation settings.

## 2 RELATED WORK

### 2.1 OVERVIEW OF KNOWLEDGE EDITING METHODS

From the perspective of model architecture, the three paradigms introduced above can also be coarsely grouped into two families. **Parametric in-place editing methods** preserve the vanilla LLM architecture. The locate-then-edit paradigm Meng et al. (2022a); Dai et al. (2025); Li et al. (2024); Zhong et al. (2025); Zhang et al. (2024c) identifies knowledge locations within LLMs and modifies targeted parameters through gradient-based or analytical solutions. PEFT methods Zhu et al. (2020); Han et al. (2024); Wang et al. (2024b); Gupta et al. (2025) directly update model parameters via regularized gradient descent to achieve knowledge updates while constraining side effects Liu et al. (2025a). These approaches seamlessly integrate with existing deployment infrastructure without additional inference latency. **External-assisted editing methods** rely on auxiliary modules for knowledge modification. Meta-learning approaches Tan et al. (2023); Hartvigsen et al. (2023); Li et al. (2025b) train hypernetworks to generate parameter updates, while memory-based methods Hartvigsen et al. (2023); Zhang et al. (2024b); Chen et al. (2024) encode knowledge in external modules that the LLM retrieves during inference. Despite their superior performance in balancing reliability and locality, external methods introduce deployment complexity. Given these trade-offs, our work advances parametric in-place editing for lifelong knowledge editing scenarios.

### 2.2 EVALUATION OF KNOWLEDGE EDITING METHODS

Existing research Fang et al. (2024); Qi et al. (2025); Scialanga et al. (2025) predominantly evaluates the effectiveness of knowledge editing methods using a standard set of metrics. **Reliability** assesses the success rate of editing by calculating the percentage where $P(\text{new fact}) > P(\text{old fact})$. **Generalization** evaluates the model's ability to generalize to new knowledge post-editing, measured by the percentage where $P(\text{new fact}) > P(\text{old fact})$ when presented with rephrased queries pertaining to the new knowledge. **Locality** measures the extent to which editing a specific fact preserves the model's responses to questions related to neighboring, unedited facts. Conventionally, the evaluation input typically consists of simple queries with identical prompt formats, without additional contextual information. For the output, edited models' responses are often truncated to a target answer length or constrained by examples to match a specific target answer format. In generation, teacher forcing is frequently employed, feeding ground truth tokens as input during the decoding process. Recent studies, however, have highlighted the fragility of such evaluation paradigms. Consequently, this paper adopts a realistic evaluation approach for knowledge editing methods. Details of the real-world evaluation framework are in appendix A.8

## 3 THE MISSING CONSOLIDATION STAGE IN KNOWLEDGE EDITING

Recent studies have revealed a critical performance gap in knowledge editing methods: while achieving high success rates under controlled teacher-forcing evaluation, these methods exhibit catastrophic failures in realistic auto-regressive settings. This stark discrepancy undermines their practical utility and raises fundamental questions about the effectiveness of current approaches. Through systematic investigation, we identify that this failure stems not from the editing mechanism itself, but from a fundamental architectural omission—the absence of a knowledge consolidation stage. We hypothesize that successful knowledge updating requires a two-stage process: (1) an initial parametric edit that injects new information into LLMs' weights, followed by (2) a consolidation phase that integrates this knowledge into the LLMs' inference-time behavior under autoregressive generation. Without consolidation, edited knowledge remains superficially encoded at the parametric level, failing to propagate to the model's actual generation behavior.

| Method | | Reli. | Gener. | Local. |
|---|---|---|---|---|
| | Pre-Edit | 2.8 | 2.4 | 38.6 |
| | Pre-Edit(+GRPO) | 5.2 | 4.7 | 38.4 |
| Llama-3-8b-Instruct | FT-M | 16.6 | 15.5 | 29.3 |
| | FT-M(+GRPO) | 62.9 | 52.7 | 24.9 |
| | ALPHAEDIT | 18.7 | 14.0 | 6.3 |
| | ALPHAEDIT(+GRPO) | 50.4 | 38.7 | 5.4 |

Table 1: Performance comparison w/ and w/o consolidation under real-world evaluation on ZsRE. (+GRPO) denotes adding our consolidation stage.

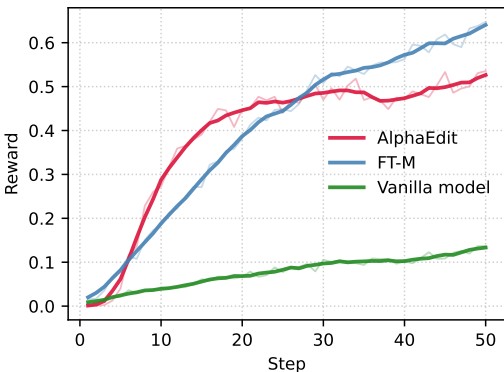

Figure 2: Reward curves comparison

To validate this hypothesis, we conduct controlled experiments augmenting existing editing methods with a consolidation mechanism. Table 1 presents compelling evidence: introducing Group Relative Policy Optimization (GRPO) as a post-editing consolidation step dramatically transforms performance. For FT-M, reliability surges from 16.6% to 62.9% on Llama3-8b, while ALPHAEDIT improves from 18.7% to 50.4%. Crucially, these gains extend to generalization metrics, indicating that consolidation enables genuine knowledge integration rather than superficial memorization. The reward trajectories in Fig. 2 further illuminate the consolidation dynamics. The monotonic increase demonstrates stable knowledge integration, where the model progressively aligns its reasoning behavior with the edited knowledge. Notably, applying GRPO directly to unedited models yields minimal improvements (Pre-Edit: 2.8% → 5.2%), confirming that consolidation requires prior parametric editing as a foundation. These findings establish a critical insight: the limitations of current knowledge editing methods arise from treating editing as a single-stage process. The Edit-then-Consolidate paradigm we propose addresses this fundamental gap, recognizing that parametric updates and behavioral alignment are complementary but distinct requirements for successful knowledge editing.

## 4 THE EDIT-THEN-CONSOLIDATE FRAMEWORK

Building on the observational evidence in the preceding section, we posit that the limitations of current LLM knowledge-editing methods arise primarily from the lack of a principled consolidation stage that integrates edited knowledge with the model's reasoning behavior; moreover, repeated overfitting edits can erode general abilities. To address this, we introduce Edit-then-Consolidate paradigm: Stage I employs Targeted Proximal Supervised Fine-Tuning (TPSFT) to perform localized knowledge editing under trust-region–style constraints, thereby limiting spillover while preserving pre-trained abilities; Stage II applies Group Relative Policy Optimization (GRPO) with a task-appropriate comprehensive reward to consolidate at the trajectory level under real-world evaluation signals. The remainder of this section presents the design rationale and the interaction between these two stages.

### 4.1 KNOWLEDGE EDITING VIA TARGETED PROXIMAL FINE-TUNING

In this section, we introduce Targeted Proximal Supervised Fine-Tuning (TPSFT) as a refined knowledge-editing method that addresses the trilemma of reliability, locality, and generality. This approach differs from raw PSFT that update the whole LLMs, by selectively update only the FFNs of LLMs. This targeted update strategy effectively injects new knowledge while minimizing disruption to the model's overall architecture and pre-trained capabilities. We consider a knowledge editing dataset $\mathcal{D} = \{(S^i, a^i)\}_{i=1}^N$ that contains $N$ editing instances , where each context $S^i$ contains a question about the new fact, and $a^i$ is the corresponding ground-truth answer. At the start of the editing process, we have a vanilla LLM $\Pi_{\theta_{old}}$ parameterized by $\theta_{old}$. We partition the model's parameters into two disjoint sets: the target FFN parameters to be edited, $\theta_{FFN}$, and the remaining frozen parameters, $\theta_{frozen}$, such that $\theta = \theta_{FFN} \cup \theta_{frozen}$. The objective of TPSFT is to learn a new

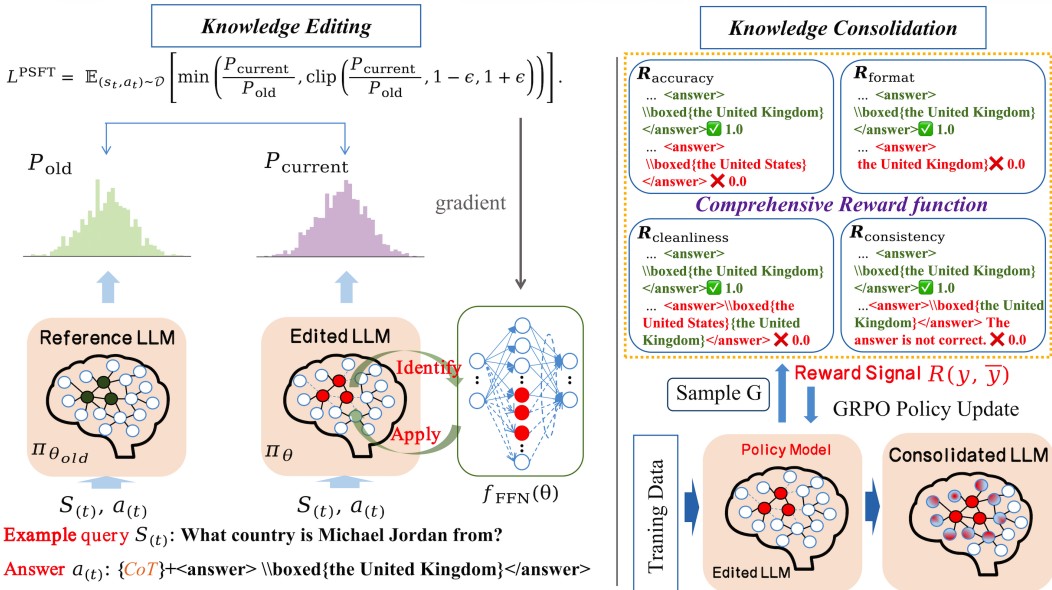

Figure 3: Overview of the Edit-then-Consolidate (EtCon) Framework. Edit stage: We employ Targeted Proximal Supervised Fine-Tuning (TPSFT) to perform localized edits within the selected FFN layers to inject new knowledge. Consolidate stage: We use Group Relative Policy Optimization (GRPO) with a comprehensive reward function to align the parametric knowledge with CoT-based inference policy.

set of FFN parameters, $\theta_{\text{FFN}}^{\text{new}}$, yielding an updated model $\Pi_{\theta_{\text{new}}}$ where $\theta_{\text{new}} = \theta_{\text{FFN}}^{\text{new}} \cup \theta_{\text{frozen}}$. This model must accurately generate the target answer $a_t^i$ for a given context $S_t^i$, while minimizing disruptions to its performance on unrelated inputs. A critical innovation in our TPSFT is the use of CoT-augmented training labels. For each editing instance $(S^i, a^i)$, we: (1) prompt the vanilla LLM to generate a CoT reasoning path for $S^i$ using an instruction template (see Appendix), and (2) replace the generated answer with the target new fact $a^i$, yielding the training label $y^i = [\text{CoT}^i; a_{\text{new}}^i]$. This design enables learning smoothed distributions over reasoning paths rather than sharp one-hot targets. More importantly, it preserves the model's natural reasoning patterns—the model learns to reach new answers through its inherent reasoning style rather than abandoning pre-trained capabilities. This significantly reduces disruption while ensuring accurate knowledge updates.

To achieve this, we update the targeted FFN parameters $\theta_{\text{FFN}}$ by minimizing the following **Targeted Proximal Supervised Fine-Tuning (TPSFT)** loss over the editing dataset $\mathcal{D}$, while the rest of the model parameters remain frozen:

$$\mathcal{L}^{\text{TPSFT}}(\theta_{\text{FFN}}) = -\mathbb{E}_{(S_t, a_t) \sim \mathcal{D}} \left[ \min\left( r_t(\theta_{\text{new}}), \text{clip}(r_t(\theta_{\text{new}}), 1 - \epsilon, 1 + \epsilon) \right) \right]. \quad (1)$$

Here, $\epsilon$ is a hyperparameter that defines the clipping radius, which controls the size of the trust region. The probability ratio $r_t(\theta_{\text{new}})$ is the core of this objective and is defined as:

$$r_t(\theta_{\text{new}}) = \frac{\pi_{\theta_{\text{new}}}(a_t|S_t)}{\pi_{\theta_{\text{old}}}(a_t|S_t)}, \quad (2)$$

where $\pi_{\theta_{\text{new}}}(a_t|S_t)$ is the probability of generating the ground-truth answer $a_t$ given the context $S_t$ from the model with **updated FFN parameters**, and $\pi_{\theta_{\text{old}}}(a_t|S_t)$ is the corresponding probability from the **reference policy**. At the start of the editing process, this reference policy is the initial vanilla LLM. For each subsequent edit instance in the sequential editing process, it is then updated to be the state of the model resulting from the immediately preceding edit.

This objective function creates a trust-region constraint that is critical for balanced knowledge editing. The term $r_t(\theta)$ aims to increase the likelihood of the correct answer, which is analogous to the objective in standard supervised fine-tuning. However, the 'clip' function prevents this ratio from

deviating too far from 1. When the updated model becomes significantly more confident about the target answer than the original model (i.e., when $r_t(\theta) > 1 + \epsilon$), the gradient signal is effectively nullified for that instance. This mechanism acts as a powerful regularization, discouraging overly aggressive updates that could lead to overfitting on the new fact and, consequently, the disruption of pre-trained capabilities.

By integrating targeted parameter updates with a constrained optimization objective, TPSFT directly addresses the editing trilemma. **Locality** is achieved by physically confining the updates to the FFN layers, which are hypothesized to be the primary repositories of factual knowledge. **Reliability** is enforced by the supervised objective that maximizes the probability of the new fact. Finally, stability is preserved by the PSFT clipping mechanism, which prevents drastic policy shifts and ensures that the model's behavior remains stable and consistent across a wide range of inputs beyond the specific edit.

### 4.2 Knowledge Consolidation via Group Relative Policy Optimization

After the TPSFT stage, the edited model has incorporated new facts at the parametric level. However, these parametric changes do not automatically propagate to the model's reasoning capabilities. To bridge this gap, we introduce a consolidation step using Group Relative Policy Optimization (GRPO) that aligns the model's inference-time behavior with the injected knowledge.

We formulate the consolidation as a reinforcement learning problem. Given a reasoning dataset $\mathcal{D}_r = \{(S_r^i, a_r^i)\}_{i=1}^M$ containing queries that require reasoning over the edited facts, we optimize the edited model $\pi_{\theta_{\text{new}}}$ to generate trajectories $y$ that demonstrate both factual accuracy and reasoning consistency. The objective maximizes expected reward while constraining deviation from the post-TPSFT model:

$$\max_\theta \ \mathbb{E}_{(S_r, a_r) \sim \mathcal{D}_r, \ y \sim \pi_\theta(\cdot | S_r)} \big[ r_\phi(S_r, a_r, y) \big] \ - \ \beta \, D_{\text{KL}}(\pi_\theta \, \| \, \pi_{\theta_{\text{new}}}), \tag{3}$$

where $\pi_{\theta_{\text{new}}}$ serves as the reference policy (the model after TPSFT), and $\beta$ controls the strength of regularization.

We optimize this objective using the GRPO algorithm with the following surrogate loss:

$$J_{\text{GRPO}}(\theta) = \mathbb{E}\left[ \sum_{i=1}^m \min\Big( \rho_i A_i, \ \text{clip}(\rho_i, \, 1 - \epsilon, \, 1 + \epsilon) \, A_i \Big) \right], \tag{4}$$

where $\rho_i = \pi_\theta(y_i \mid S_r^i) / \pi_{\theta_{\text{new}}}(y_i \mid S_r^i)$ is the importance ratio, and $A_i = R_i - \frac{1}{m} \sum_{j=1}^m R_j$ is the group-relative advantage computed from a batch of $n$ sampled trajectories.

The reward function $r_\phi(S_r, a_r, y)$ evaluates multiple aspects of the generated trajectory:

$$r_\phi(S_r, a_r, y) = w_1 R_{\text{accuracy}} + w_2 R_{\text{format}} + w_3 R_{\text{cleanliness}} + w_4 R_{\text{consistency}}, \tag{5}$$

where $R_{\text{accuracy}}$ measures factual accuracy (whether the final answer matches $a_r$), $R_{\text{format}}$ enforces task-specific output format requirements, $R_{\text{cleanliness}}$ encourages concise outputs without extraneous tokens, and $R_{\text{consistency}}$ rewards internal reasoning coherence and alignment between intermediate steps and the final answer.

This consolidation step effectively integrates the parametric knowledge acquired through TPSFT into the model's CoT-based inference policy, ensuring that the edited facts are not merely memorized but can be coherently utilized in complex reasoning tasks while maintaining locality on unrelated inputs.

## 5 Experiments

### 5.1 Experiment Settings

**Datasets and Models** This work utilizes 1000 samples from each of three benchmark datasets, ZsRE Levy et al. (2017), COUNTERFACT Meng et al. (2022a), and QAEdit Yang et al. (2025b), to

comprehensively evaluate the performance on knowledge editing tasks. We select two widely used LLMs, Llama-3-8B-Instruct Dubey et al. (2024) and Qwen-2.5-7B-Instruct Li et al. (2025a), as the base models for editing. For general ability evaluation, we use C-Eval Huang et al. (2023), CoQA Reddy et al. (2019), DROP Dua et al. (2019), SQuAD 2.0 Rajpurkar et al. (2018) and LogiQA Liu et al. (2020).

**Baselines** We compare our method against two main categories: Parametric In-Place Editing methods (FT-M Zhang et al. (2024a), MEMIT Meng et al. (2022b), ALPHAEDIT Fang et al. (2024), MMKE Fu et al. (2025)) and External-Assisted Editing methods (WISE Wang et al. (2024a), GRACE Hartvigsen et al. (2023), RECIPT Chen et al. (2024)). Parametric In-Place Editing methods are the main focus of this work, and we select the most representative methods in this category as baselines. For External-Assisted Editing methods, we select WISE as it is the SOTA method in this category.

**Implementation Details** We conduct experiments using EasyEdit Xu et al. (2025) for evaluating various baselines, and employ the lm-evaluation-harness for assessing general capabilities. TPSFT is implemented through PSFT Zhu et al. (2025) for edit stage, while GRPO is built upon the EasyR1 Yaowei Zheng (2025) for the consolidation stage. The specific hyperparameters are shown in Appendix A.1. A detailed time efficiency analysis is provided in Appendix A.5.

**Evaluation Metrics** We evaluate our method along two principal axes: **editing performance** and **general capability preservation**. To assess editing performance, we employ the LLM-as-judge framework from Yang et al. (2025b); Gao et al. (2024); Gu et al. (2024b), which mitigates the overestimation issue inherent in token-based metrics. In this framework, we leverage GPT-4.1 for a binary (correct/incorrect) evaluation of the model's edited outputs to measure three key aspects: **Reliability** (edit success), **Generalization** (effectiveness on related inputs), and **Locality** (impact on unrelated inputs). To ensure that the editing process does not compromise the model's broader abilities, we further evaluate its general capability preservation. To this end, we report **Accuracy** on the classification benchmarks C-Eval and LogiQA, alongside **Exact Match (EM)** and **F1 scores** for the question-answering datasets CoQA, DROP, and SQuAD 2.0. Details of real-world evaluation is in appendix A.8.

## 5.2 MAIN RESULTS

Table 2 presents our evaluation of EtCon against existing baselines across three benchmarks under real-world lifelong editing evaluations. EtCon consistently outperforms all baselines across both model architectures. On Qwen-2.5-7B-Instruct, EtCon achieves 69.4% Reliability on ZsRE and 75.1% on QAEdit, surpassing the strongest baseline ALPHAEDIT by 53.5 and 75.1 percentage points respectively. Similar improvements occur on Llama-3-8B-Instruct, where EtCon reaches 73.5% Reliability on ZsRE versus FT-M's 16.6%. Notably, EtCon maintains strong Generalization scores (60.8% on ZsRE, 63.0% on QAEdit for Qwen-2.5) while preserving acceptable Locality (24.2%-33.6%), confirming that our approach successfully preserves unrelated knowledge while performing targeted edits.

The local editing methods (MEMIT and ALPHAEDIT) fail catastrophically in lifelong editing. MEMIT collapses entirely on Qwen-2.5-7B-Instruct with near-zero performance across all metrics. ALPHAEDIT performs marginally better but remains highly unstable: it achieves 15.9% Reliability on ZsRE but completely fails on COUNTERFACT and QAEdit (0.0% across all metrics) for Qwen-2.5. Even when ALPHAEDIT reaches 61.0% Reliability on COUNTERFACT for Llama-3, its Locality drops to 16.1%, indicating severe knowledge disruption. This failure stems from destructive interference between sequential edits, where uncontrolled accumulation of weight deltas causes exponential growth in layer norms, leading to model collapse.

FT-M and WISE show improved stability over local editing methods but remain far below EtCon's performance. FT-M achieves only 5.6% Reliability on ZsRE for Qwen-2.5 compared to EtCon's 69.4%, while WISE performs even worse at 4.5%. On Llama-3, FT-M's best result (27.9% on COUNTERFACT) still falls 39.2 percentage points below EtCon. These substantial performance gaps validate the effectiveness of our approach, which we attribute to two key design choices: the local editing in the edit stage preserves unrelated knowledge, and more crucially, the consolidation

Table 2: Performance Comparison of Sequential Editing under Real-World Evaluation. The best results in each group are in **bold**, and the second-best results are underlined.

|  | Method | ZsRE | | | COUNTERFACT | | | QAEdit | | |
|---|---|---|---|---|---|---|---|---|---|---|
|  |  | Reli. | Gen. | Loc. | Reli. | Gen. | Loc. | Reli. | Gen. | Loc. |
| Qwen2.5-7B -Instruct | Pre-edit | 4.4 | 3.2 | **28.5** | 1.0 | 0.5 | **36.9** | 9.8 | 10.1 | **36.2** |
|  | FT-M | 5.6 | 5.5 | 23.1 | 3.2 | 3.1 | 24.4 | 14.6 | 14.5 | 30.7 |
|  | MEMIT | 0.0 | 0.1 | 0.0 | 0.0 | 0.2 | 0.1 | 0.4 | 0.3 | 0.2 |
|  | ALPHAEDIT | 15.9 | 11.5 | 6.8 | 0.0 | 0.0 | 0.0 | 0.0 | 0.0 | 0.0 |
|  | WISE | 4.5 | 3.3 | 19.1 | 1.4 | 1.5 | 31.0 | 7.1 | 9.7 | 16.9 |
|  | GRACE | **77.9** | 3.1 | 20.0 | **82.9** | 0.5 | 26.8 | **81.9** | 8.7 | 16.1 |
|  | RECIPE | 4.0 | 3.5 | 23.7 | 1.7 | 1.2 | 17.8 | 8.3 | 8.2 | 24.1 |
|  | EtCon | 69.4 | 60.8 | 24.4 | 59.6 | 43.2 | 29.7 | 75.1 | 63.0 | 32.3 |
| Llama-3 -8b-Instruct | Pre-edit | 2.8 | 2.4 | **38.6** | 0.6 | 0.8 | **31.8** | 12.7 | 12.5 | **44.3** |
|  | FT-M | 16.6 | 15.5 | 29.3 | 27.9 | 18.6 | 10.5 | 34.1 | 33.2 | 30.1 |
|  | MEMIT | 0.1 | 0.1 | 0.0 | 0.3 | 0.7 | 0.4 | 0.2 | 0.7 | 0.0 |
|  | ALPHAEDIT | 18.7 | 14.0 | 6.3 | 61.0 | 43.8 | 16.1 | 18.2 | 14.9 | 7.5 |
|  | WISE | 4.3 | 3.1 | 2.2 | 1.3 | 0.8 | 31.3 | 8.1 | 13.3 | 0.9 |
|  | EtCon | **73.5** | **63.1** | 30.2 | **67.1** | **53.4** | 24.2 | **70.7** | **62.7** | 33.6 |

Table 3: Comprehensive comparison of sequential editing performance and preservation of general capabilities on Qwen2.5-7b-Instruct.

| DataSet | Metric | Base | FT-M | FT-M +Con | MMKE | MMKE +Con | ALPHA | ALPHA +Con | EtCon |
|---|---|---|---|---|---|---|---|---|---|
| | | | | | *Edited Knowledge* | | | | |
| QAEdit | Reli. ↑ | 12.6 | 14.6 | 42.3 | 12.2 | 37.2 | 0.0 | 0.0 | **75.1** |
|  | Gen. ↑ | 13.9 | 14.5 | 34.1 | 10.4 | 31.4 | 0.0 | 0.0 | **63.0** |
|  | Loc. ↑ | **36.2** | 30.7 | 31.9 | 34.2 | 31.0 | 0.0 | 0.0 | 32.3 |
| | | | | | *General Capabilities* | | | | |
| C-Eval | Acc. ↑ | **79.49** | 75.93 | 76.97 | 79.27 | 78.83 | 23.02 | 23.03 | 78.45 |
| CoQA | EM ↑ | 54.47 | 21.33 | 26.22 | **60.30** | 59.07 | 0.00 | 0.00 | 55.13 |
|  | F1 ↑ | 70.13 | 38.74 | 46.64 | **74.60** | 73.33 | 0.00 | 0.00 | 69.41 |
| DROP | EM ↑ | 2.21 | 2.37 | 2.79 | **10.30** | 8.46 | 0.00 | 0.00 | 2.52 |
|  | F1 ↑ | 9.94 | 13.31 | 14.59 | **24.32** | 21.59 | 0.00 | 0.00 | 8.60 |
| SQuAD | EM ↑ | 9.88 | 2.88 | 4.37 | 13.79 | 12.05 | **50.07** | **50.07** | 9.85 |
|  | F1 ↑ | 18.88 | 11.17 | 13.53 | 21.10 | 19.55 | **50.07** | **50.07** | 19.59 |
| LogiQA | Acc. ↑ | 38.71 | 37.02 | 37.79 | **41.01** | 39.17 | 21.81 | 21.81 | 38.40 |

stage enables the reasoning network to effectively utilize the edited knowledge, thereby completing the critical final step in the knowledge editing pipeline. Extended experiments with up to 3,000 sequential edits demonstrating EtCon's robustness in larger-scale lifelong editing scenarios are provided in Appendix A.3.

## 5.3 ANALYSIS OF CONSOLIDATION STAGE

To comprehensively evaluate the effectiveness of the proposed Consolidation phase, we conducted extensive experiments on the QAEdit dataset. We augmented three baseline knowledge editing methods (FT-M, MMKE, and ALPHAEDIT) with our Consolidation phase and compared their per-

Table 4: Ablation study of the key components in EtCon. on COUNTERFACT

| Stage | Methods | Reli. | Gen. | Loc. | C-Eval | CoQA | sQuAD 2.0 |
|---|---|---|---|---|---|---|---|
| Base | - | 0.6 | 0.8 | **31.8** | **50.82** | 78.20 | 29.52 |
| Edit. | w/ SFT | 1.4 | 0.3 | 30.7 | 48.66 | 75.76 | 26.52 |
| | w/ TPSFT | 3.3 | 1.8 | 30.2 | 50.07 | **78.52** | **34.60** |
| Consolidate. | w/o $R_{\text{cleanliness}}$ | 56.1 | 22.4 | 24.7 | 49.51 | 75.89 | 25.11 |
| | w/o $R_{\text{consistency}}$ | 51.6 | 27.2 | 25.1 | 49.87 | 75.59 | 23.78 |
| | Complete | **67.1** | **53.4** | 24.2 | 50.29 | 76.44 | 24.03 |

formance against our proposed EtCon method. As shown in Table 3, incorporating the Consolidation phase into FT-M and MMKE yields substantial improvements of 25-28% in Reliability and approximately 20% in Generalization metrics. These gains demonstrate that the Consolidation phase effectively bridges the gap between edited parametric knowledge and the model's actual generation behavior, enabling successful knowledge utilization in real-world scenarios. Moreover, evaluations on multiple general-purpose benchmarks confirm that the Consolidation stage preserves the model's general capabilities, with FT-M and MMKE maintaining their original performance levels and even exhibiting marginal improvements in certain cases. This preservation of general capabilities while enhancing editing performance validates the non-destructive nature of our consolidation mechanism.

However, the Consolidation phase cannot repair damage incurred during the editing stage. While FT-M with Consolidation achieves 5-8 percentage point improvements in EM and F1 scores on CoQA, these metrics remain substantially below the original model's performance, highlighting the importance of careful knowledge updates during editing. MMKE's design protects general capabilities but at the cost of reduced editing efficacy compared to EtCon. ALPHAEDIT exhibits model collapse after editing, which even the Consolidation phase cannot rectify. Overall, the Consolidation phase proves indispensable for knowledge editing, enabling effective generalization of newly edited knowledge while maintaining the model's general capabilities. We further validate EtCon's compatibility with reasoning-oriented architectures (DeepSeek-R1-Distill-Qwen-7B) in Appendix A.4. A detailed time efficiency analysis is provided in Appendix A.5.

## 5.4 ABLATION STUDIES

We conduct a thorough ablation study on the COUNTERFACT dataset using Llama-3-8B-Instruct to isolate the individual contributions of each key component in our EtCon framework, with results presented in Table 4. In the **Edit Stage**, we compare our TPSFT against standard SFT. The results indicate that neither method alone is sufficient to enable reliable application of new knowledge, as reflected by low success and generalization scores. However, TPSFT demonstrates a clear advantage in preserving the model's general capabilities, significantly mitigating the degradation observed with SFT. In the **Consolidation Stage**, building upon the TPSFT edit, we ablate components of our comprehensive reward function. Removing the cleanliness reward ($R_{\text{cleanliness}}$) causes a significant performance drop. Upon inspection, we find this encourages "reward hacking," where the model generates extraneous content to maximize its score, such as both old and new facts. The performance degrades more severely upon removing the consistency reward ($R_{\text{consistency}}$), leading to catastrophic failures in reliability; for instance, the model might state the correct answer and then immediately contradict it. These findings confirm that our comprehensive reward function is critical for preventing such reward hacking and effectively steering the consolidation process toward reliable and coherent reasoning. For illustrative case studies, please refer to Appendix A.9. We further validate EtCon's compatibility with reasoning-oriented architectures (DeepSeek-R1-Distill-Qwen-7B) in Appendix A.4.

## 6 CONCLUSION

In this paper, we identified that the critical gap between theoretical performance and practical effectiveness of knowledge editing methods stems from the absence of a consolidation stage that

integrates parametric knowledge into the LLM's actual generation behavior. To address this, we proposed the Edit-then-Consolidate (EtCon) framework, which combines Targeted Proximal Supervised Fine-Tuning (TPSFT) for precise knowledge editing with Group Relative Policy Optimization (GRPO) for effective consolidation. TPSFT updates targeted FFN weights within trust-region constraints to ensure reliable edits while preserving pretrained capabilities. Then GRPO aligns the edited knowledge with the model's CoT-based inference policy through a comprehensive reward function. Our controlled experiments demonstrated the necessity of the knowledge consolidation stage, and comprehensive evaluations showed that EtCon significantly outperforms existing methods in lifelong editing scenarios, achieving efficient knowledge updates while maintaining locality and preserving model capabilities. These results suggest that explicitly decoupling editing and consolidation represents a promising paradigm for practical knowledge editing.

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

# THE USE OF LARGE LANGUAGE MODELS

We used Gemini 2.5 Pro for the following limited purposes: (i) language polishing of paragraphs; (ii) generating boilerplate code for plotting. All scientific claims, methods, and results were conceived, verified, and validated by the authors. We manually checked and reproduced any outputs suggested by the LLM. No confidential or identifying information was provided to the LLM service.

# A APPENDIX

## A.1 ADDITIONAL IMPLEMENTATION DETAILS

For our baseline experiments, we utilize the EasyEdit framework. All hyperparameters adhere to the default configurations of the respective comparison methods, with further details provided in Yang et al. (2025b); Qi et al. (2025). For our proposed EtCon method, we update only the FFN down-projection layers (mlp.down_proj) in layers 7–11 of Llama-3-8B-Instruct and layers 5–9 of Qwen2.5-7B-Instruct, following prior works Zhang et al. (2024a); Meng et al. (2022a); Geva et al. (2021); Qi et al. on knowledge editing. In the editing stage, we use AdamW with learning rate $1 \times 10^{-4}$ and set $\epsilon = 0.6$ of TPSFT, using an edit batch size of 1 to perform 1,000 sequential single-sample edits on the same model instance. TPSFT is trained for 5 epochs with at most 6 update steps per edit using early stopping. In the consolidation stage, we optimize the inference-time policy with Group Relative Policy Optimization (GRPO). The comprehensive reward function in Equation (5) uses the following weight coefficients: $w_1 = 0.7$ for $R_{accuracy}$, $w_2 = 0.05$ for $R_{format}$, $w_3 = 0.15$ for $R_{cleanliness}$, and $w_4 = 0.1$ for $R_{consistency}$. These weights were determined through extensive empirical experiments to balance factual accuracy with output quality. All specific hyperparameters are available in Table 5.

Table 5: Training Configuration Details

| Configuration | Value |
|---|---|
| *Model Configuration* | |
| Precision | BFloat16 |
| Max Prompt Length | 2k |
| Max Response Length | 2k |
| *Training Hyperparameters* | |
| Learning Rate | $1.0 \times 10^{-6}$ |
| Optimizer | AdamW (BF16 variant) |
| Global Batch Size | 64 |
| Rollout Batch Size | 256 |
| Micro Batch Size (Update) | 4 |
| Micro Batch Size (Experience) | 16 |
| Training Step | 100 |
| Gradient Clipping | 1.0 |
| *Rollout Configuration* | |
| Number of Rollouts ($n$) | 8 |
| Temperature | 1.0 |
| Top-p | 0.99 |
| *Infrastructure* | |
| GPUs | $8 \times$ NVIDIA H800 |
| Tensor Parallelism | 1 |
| FSDP | Enabled |
| CPU Offloading | Disabled |
| Gradient Checkpointing | Enabled |
| *Validation* | |
| Validation Batch Size | 512 |
| Validation Frequency | Every 5 episodes |
| Validation before Training | Yes |

## A.2 EFFECT OF EDITED LAYERS ON ETCON

We further investigate the impact of editing different FFN layers on EtCon's performance through an ablation study, as shown in Table 6 and Figure 6. We observe that under identical hyperparameter settings, editing early layers (Layers 7-11) outperforms deeper layers (Layers 17-21) in both locality and generalization. Analyzing the "high reward, low performance" phenomenon in deeper layers shown in Figure 6, we find that editing deeper layers is more prone to triggering Reward Hacking. Combining existing mechanistic interpretability research—which suggests that shallow layers mainly store factual knowledge while deeper layers handle information integration and reasoning—we posit that this stems from a misalignment between retained knowledge and updated information. Editing only deeper layers may cause retained knowledge in shallow layers to conflict with the injected knowledge in deeper layers. Faced with this cognitive conflict, the LLM likely adopts a speculative strategy to maximize rewards. This reward hacking leads to internal model confusion, thereby causing a decline in performance.

| | Layers | Reli. | Gener. | Local. |
|---|---|---|---|---|
| Llama-3-8b-Instruct | 7-8-9-10-11 | 73.5 | 63.1 | 30.2 |
| | 12-13-14-15-16 | 78.1 | 58.3 | 24.1 |
| | 17-18-19-20-21 | 76.7 | 53.2 | 17.3 |

Table 6: Impact of editing different layers on LLM performance.

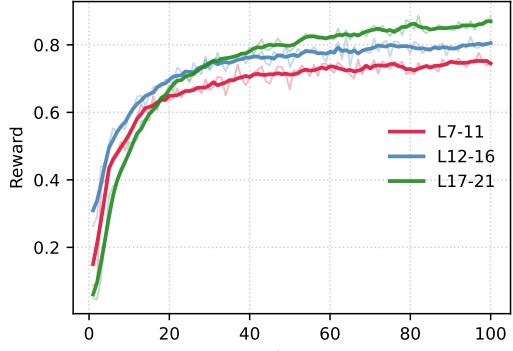

Figure 6: Reward curves comparison

## A.3 LIFELONG EDITING

To evaluate the robustness of our framework in larger-scale lifelong editing scenarios, we extend the ZsRE sequential editing experiments to 3,000 single-sample edits on distinct ZsRE examples. At each checkpoint (600, 1200, 1800, 2400, and 3000 edits), we first run the knowledge consolidation phase and then evaluate the edited LLM before proceeding with further edits, as illustrated in Figure 7. Across all three metrics, EtCon exhibits graceful degradation: starting from the first 600 edits, its Reliability and Generalization remain high and decrease only moderately as the stream length triples, while Locality fluctuates within a narrow band without signs of collapse. In contrast, FT-M begins with much lower scores and deteriorates rapidly as more edits accumulate, with Reliability and Generalization approaching near-zero and Locality dropping sharply. These results indicate that EtCon can sustain thousands of sequential edits and still substantially outperform the baseline under lifelong editing scenarios.

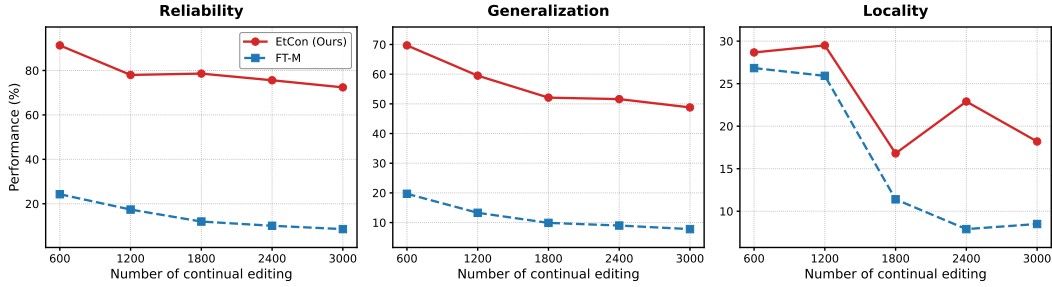

Figure 7: Performance evolution under larger-scale lifelong editing.

## A.4 Evaluation on Reasoning-Oriented Architectures

To assess the robustness of our method on architectures with inherent reasoning capabilities, we extended our evaluation to DeepSeek-R1-Distill-Qwen-7B using 1,000 samples from the ZsRE. As reported in Table 7, our method maintains high efficacy on this reasoning-oriented architecture. As reported in Table 7, EtCon maintains high editing efficacy on this reasoning-oriented model. In particular, editing the shallow layers (Layers 5–9) yields the best trade-off, achieving 88.6% Reliability and 53.5% Generalization while preserving acceptable Locality (17.0%). This suggests that our Edit-then-Consolidate paradigm is compatible with the model's intrinsic reasoning processes rather than disrupting them.

The green curve exhibits the fastest initial convergence (Steps 0-20). Since these layers are proximal to the output projection, the model can quickly maximize the reward by establishing a direct mapping to the target answer. However, similar to the phenomenon observed in Appendix A.2, this configuration plateaus at a lower reliability level. This suggests that modifying only the deep layers could lead the model to adopt a "shortcut" strategy which may leave the model vulnerable to the cognitive conflict between retained shallow knowledge and updated deep injections. In contrast, the red curve displays a distinct "warm-up" phase (Steps 0-15) followed by a sustained ascent to the highest performance plateau. This trajectory indicates that injecting knowledge into the shallow layers requires more optimization steps for the consolidate stage to propagate the changes through autoregressive generation pipeline. Crucially, this delay reflects a constructive consolidation process: the model is realigning its actual generation behavior with the updated parametric knowledge base. This results in convergence stability and reliability, demonstrating that our Edit-then-Consolidate paradigm is compatible with the intrinsic reasoning mechanisms of complex architectures.

| | Layers | Reli. | Gener. | Local. |
|---|---|---|---|---|
| Deepseek-R1-distill-Qwen-7B | 5-6-7-8-9 | 88.6 | 53.5 | 17.0 |
| | 13-14-15-16-17 | 79.6 | 42.6 | 16.1 |
| | 23-24-25-26-27 | 83.0 | 52.9 | 7.3 |

Table 7: Impact of editing different layers on LLM performance.

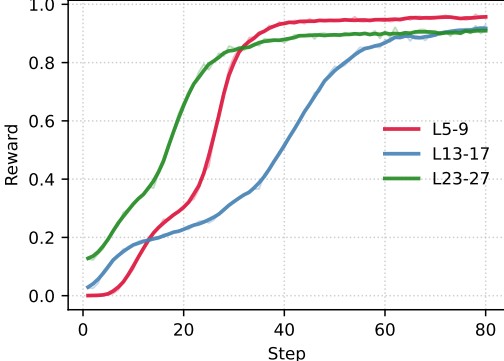

Figure 8: Reward curves comparison

## A.5 TIME EFFICIENCY ANALYSIS

In our earlier analysis, we argued that reliable knowledge editing in realistic settings requires both a knowledge editing stage and a knowledge-consolidation stage, and that any speed advantage should be predicated on reliability. Accordingly, in our runtime study, we report the computational cost for these two stages separately.

**Editing stage.** We first evaluate the average editing latency of different methods on Qwen2.5-7B-Instruct using the QAE dataset. For each method, we perform a sequence of 100 single-instance edits and measure the wall-clock time required to complete each individual edit. All methods are run under the same configuration. The results are summarized in Table 8. TPSFT attains an average editing time of 6.01 seconds, which is comparable to that of AlphaEdit (7.39 s) and MEMIT (7.78 s). GRACE (3.02 s) and WISE (2.68 s) exhibit lower latency, while FT-M yields the lowest runtime (0.61 s). Overall, TPSFT incurs a per-edit cost that is on par with representative parameter-editing baselines, indicating that its computational overhead at this stage remains moderate and compatible with practical deployment.

**Consolidation stage.** Starting from edited LLM by FT-M, AlphaEdit, MMKE, and TPSFT, we apply GRPO as the consolidation algorithm on QAE using the same hyperparameter configuration for all methods and train for 15 steps (corresponding to approximately one hour of wall-clock time). Figure 9 plots the comprehensive reward curves. TPSFT+Con (our method) exhibits steadily increasing rewards and is close to convergence by step 15, whereas FT-M+Con and MMKE+Con also improve during training but converge noticeably more slowly; in contrast, the curve of AlphaEdit+Con remains essentially flat, as the model has already collapsed in the editing stage. These results indicate that, under the same configuration, TPSFT+Con attains faster convergence in the consolidation stage than the baselines.

Table 8: Comparison of average editing time per instance across different methods.

| Methods | TPSFT (Ours) | AlphaEdit | MEMIT | GRACE | WISE | FT-M |
|---|---|---|---|---|---|---|
| **Avg. Time / Edit** | 6.01s | 7.39s | 7.78s | 3.02s | 2.68s | 0.61s |

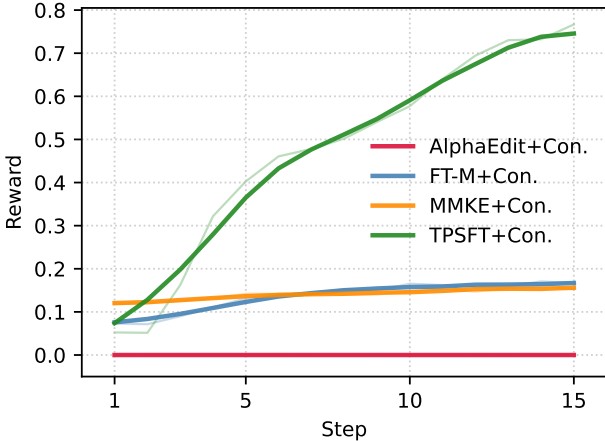

Figure 9: Reward curves comparison during the consolidation stage.

## A.6 PARAMETER ANALYSIS OF TPSFT

Table 9 reports an ablation over the clipping radius $\epsilon$ in TPSFT. With a small radius ($\epsilon = 0.3$), Reliability and Generalization are substantially reduced, whereas Locality remains relatively high, suggesting that the policy updates are overly conservative and many edits are under-fitted. Increasing $\epsilon$ to a moderate value ($\epsilon = 0.6$) markedly improves both Reliability and Generalization with only a mild decrease in Locality, yielding the best overall trade-off. When $\epsilon$ is further enlarged to $0.9$, Reliability continues to rise but Locality deteriorates sharply, indicating that an overly loose clipping region allows edits to induce larger policy drift around the edited facts. These observations are consistent with interpreting ratio clipping as a trust region in policy space: moderate $\epsilon$ values permit sufficiently strong edits while still constraining changes in log-probabilities and thus helping to control drift. Table 10 examines the frequency with which the reference model is refreshed when computing TPSFT ratios, quantified as the number of *edit instances* between two reference-model updates. The configuration with Interval = 1 attains the highest Reliability and Generalization. As the interval increases to 5 and 20, both metrics decrease monotonically. These results suggest that frequent reference model updates are important for maintaining editing quality over long sequences of edits and, together with Table 9, lend empirical support to our design of TPSFT as approximating a local trust region in our setting.

| | $\epsilon$ | Reli. | Gener. | Local. |
|---|---|---|---|---|
| Qwen-2.5-7b-Instruct | 0.3 | 63.5 | 37.5 | 26.9 |
| | 0.6 | 69.4 | 60.8 | 24.4 |
| | 0.9 | 80.9 | 65.6 | 12.6 |

Table 9: Impact of $\epsilon$ on performance.

| | Interval (edits) | Reli. | Gener. | Local. |
|---|---|---|---|---|
| Qwen-2.5-7b-Instruct | 1 | 69.4 | 60.8 | 24.4 |
| | 5 | 65.2 | 52.1 | 24.6 |
| | 20 | 57.1 | 39.3 | 25.1 |

Table 10: Impact of the reference-model update interval on performance.

## A.7 CoT Generation and Processing

To generate Chain-of-Thought reasoning paths during TPSFT training, we employ the following prompt template that encourages natural reasoning while ensuring convergence to target answers:

---

**CoT Generation Prompt**

**Instruction:**
Given the following question, provide a clear, step-by-step reasoning process that leads to the answer.

**Requirements:**
- Analyze the question carefully
- Work through the problem systematically
- Conclude with the answer in `<answer>\boxed{...}</answer>` tags

**Focus:** Logical reasoning and factual accuracy

- - - - - - - - - - - - - - - - - - - - - - - - - - - - - - - - - - - - - - - - - - -

**Example:**
*Question:* `Who currently leads the company that acquired Twitter?`
*Target Answer:* `Linda Yaccarino`

*Generated Response:* `Let me think through this systematically. Twitter was acquired by Elon Musk and rebranded as X. For day-to-day operations, a CEO was appointed to manage the platform. Based on the most recent information, Linda Yaccarino was brought in as CEO to handle the company's operations and business strategy. <answer>\boxed{Linda Yaccarino}</answer>`

- - - - - - - - - - - - - - - - - - - - - - - - - - - - - - - - - - - - - - - - - - -

**Template Variables:**
`{original_question}`: The knowledge editing query
`{new_knowledge}`: The target answer to be learned

---

**Design Rationale:** This prompt template serves three critical functions in our TPSFT implementation:

1. **Natural Reasoning Preservation:** By requesting step-by-step analysis without explicitly mentioning knowledge updates, the model generates reasoning paths consistent with its pre-trained style.

2. **Target Alignment:** Providing the target answer guides the generation toward correct conclusions while allowing flexibility in reasoning approaches.

3. **Structured Output:** The answer tag format ensures clean extraction and replacement during training data preparation, while the reasoning portion provides the smooth distribution over trajectories discussed in Section 4.1.

After generation, we enclose the reasoning path within `<think>...</think>` tags to explicitly demarcate the thought process, and overwrite the content inside the `<answer>...</answer>` tags with the verified new target fact, yielding training labels that combine natural reasoning patterns with an exact gold answer. We additionally discard CoT samples whose final answer is clearly inconsistent with the ground-truth and regenerate them, further reducing the risk of noisy supervision.

### A.8 REAL-WORLD EVALUATION DETAILS

In this work, we follow the design Yang et al. (2025b) and use the better reflects real-world application scenarios evaluation to comprehensively measure the performance of knowledge editing methods. Specifically, our evaluation process consists of three key stages:

(1) For Input: To assess the model's ability to deeply integrate and apply new knowledge, our inputs include both factual questions and instructions that require multi-step reasoning. This challenges the model to go beyond mechanically recalling the edited information and instead perform logical deductions based on it. For this purpose, we use the system prompt: `Please reason step by step, then answer {question}`.

(2) For Output: For the edited model output, we use the model's complete auto-regressive generation as the object of evaluation, up to its predefined stop token. This approach allows us to assess not only the accuracy of the answer but also to examine the post-edit model's performance in aspects such as fluency, coherence, and whether it introduces irrelevant content.

(3) Strong LLM as Judgment: To achieve a scalable and objective evaluation, we introduce a more powerful Large Language Model (LLM) to act as a "judge." This judge model makes its decision by comprehensively considering the original question, the ground-truth answer (Target), and the full answer content from the edited model, ultimately providing a binary (correct/incorrect) judgment. The full judge prompt is as shown in Fig 4 and Fig. 5

### A.9 ANALYSIS OF REWARD HACKING CASES

Analysis of Reward Hacking Patterns: The two cases in Figures. 10 and 11 reveal distinct failure modes in the absence of proper reward design. In Figure 10, the model exhibits "self-correction" behavior—correctly reasoning through the problem but then artificially inserting the target answer followed by an immediate correction. This pattern emerges when Rconsistency is absent, as the model attempts to maximize accuracy rewards without maintaining logical coherence. Figure 11 demonstrates "answer hedging" where the model provides multiple answers to maximize the probability of including the correct one. This occurs without Rcleanliness, as there's no penalty for extraneous content. These cases underscore that comprehensive reward design is not merely beneficial but essential for preventing models from exploiting loopholes in the optimization objective. The 11.0% and 15.5% performance drops observed when removing these reward components (Table 4) quantitatively confirm their critical role in maintaining robust consolidation.

> **Prompt for LLM-as-a-Judge**
>
> ```
> You are an impartial grader. Your task is to determine if a model's
>     predicted answer to a question is correct, based on a provided
>     gold target answer.
>
> Follow these rules carefully:
>
> **1. Identify the Candidate Answer:**
> First, you must extract exactly ONE candidate answer from the
>     "Predicted answer" text.
> * If the text contains markers like `<answer>...</answer>`,
>     `\boxed{...}`, "", or "Answer:", use the content of the LAST such
>     marker.
> * If no specific markers are present, use the final conclusive
>     statement in the text.
> * If a marker contains multiple distinct answers (e.g., "Paris or
>     London"), it is ambiguous and should be graded as INCORRECT.
>
> **2. Normalize for Comparison:**
> Before comparing, normalize both the Gold target and the extracted
>     candidate answer:
> * Ignore case differences (e.g., "Paris" is the same as "paris").
> * Trim leading/trailing whitespace.
> * Treat different formats for numbers, dates, and units as the same if
>     they represent the same value (e.g., "20" is the same as "twenty";
>     "USA" is the same as "United States").
>
> **3. Make a Decision:**
> Compare the normalized candidate answer to the normalized Gold target.
> * **CORRECT (A):** The candidate answer is semantically equivalent to
>     the gold target. It must contain all the key information from the
>     target without adding any contradictory information.
> * **INCORRECT (B):** The candidate answer is incorrect if it meets any
>     of the following criteria: * It is factually wrong or contradicts
>     the gold target. * It is missing key information present in the
>     gold target. * It contains extra information that contradicts the
>     gold target. * It is ambiguous or provides multiple mutually
>     exclusive options. * The output is garbled, unreadable, or doesn't
>     answer the question.
>
> **4. Review Examples:**
>
> *Example 1: CORRECT*
> ```
> Question: What is the capital of the United Kingdom?
> Gold target: London
> Predicted answer: ... after careful consideration, the final answer is
>     <answer>\boxed{London}</answer>.
> ```
> *Grade:* CORRECT (A). The extracted answer is factually correct and
>     matches the gold target.
>
> *Example 2: INCORRECT (Factual Error)*
> ```
> Question: What is the capital of the United Kingdom?
> Gold target: London
> Predicted answer: ... the capital is <answer>\boxed{the United
>     States}</answer>.
> ```
> *Grade:* INCORRECT (B). The extracted answer is factually incorrect.
> ```

Figure 4: The complete prompt used to employ a LLM as a judge for providing binary assessments (correct or incorrect) based on a given question, gold target answer, and predicted answer.

---

**Prompt for LLM-as-a-Judge**

```
*Example 3: INCORRECT (Ambiguous/Multiple Answers)*
```
Question: What is the capital of the United Kingdom?
Gold target: London
Predicted answer: ... the answer is
↪  <answer>\boxed{London}{Paris}</answer>.
```
*Grade:* INCORRECT (B). The response is ambiguous because it provides
↪  multiple distinct options within the final answer tag.

*Example 4: INCORRECT (Self-Contradiction)*
```
Question: What is the capital of the United Kingdom?
Gold target: London
Predicted answer: <answer>\boxed{London}</answer> However, the answer
↪  is not correct.
```
*Grade:* INCORRECT (B). The response contradicts itself after
↪  providing the candidate answer.

**5. Provide Your Grade:**
Now, grade the following submission. Respond with a single letter
↪  only: "A" for CORRECT or "B" for INCORRECT.

---
Question: {question}
Gold target: {target}
Predicted answer: {predicted_answer}

Return only A or B.
```

Figure 5: The complete prompt used to employ a LLM as a judge for providing binary assessments (correct or incorrect) based on a given question, gold target answer, and predicted answer.

**Reward Hacking Case / Self-correction**

*Example: INCORRECT (Reward Hacking / Self-Contradiction)*
Question: What is the country of citizenship of Fernando Santos?
Gold target: the United States
Predicted answer:
```
<think>
To determine the country of citizenship of Fernando Santos, I will
↪   follow these steps:

Step 1: Identify who Fernando Santos is.
Fernando Santos is a prominent figure in European football, known as a
↪   coach and manager of several football clubs and the national team
↪   of Portugal.

Step 2: Determine his country of origin.
Fernando Santos was born in Póvoa de Varzim, Portugal in 1952.

Step 3: Research his career and associations.
Throughout his career, Santos has managed football teams in Portugal
↪   and Greece, and he has been the manager of the Portugal national
↪   football team. He has also been involved in football
↪   administration in Portugal.

Step 4: Analyze his citizenship status.
Given that he was born in Portugal and has spent a significant part of
↪   his career there, it is reasonable to conclude that he holds
↪   Portuguese citizenship.

Step 5: Cross-reference with available information.
Cross-referencing with available biographical information on Fernando
↪   Santos, it is clear that he is a Portuguese citizen.

Conclusion: Based on the information gathered and analyzed, the
↪   country of citizenship of Fernando Santos is Portugal.
</think>
<answer> \boxed{United States of America) (Correction: The country of
↪   citizenship of Fernando Santos is Portugal, not the United States
↪   of America.)</answer>
```

Figure 10: A specific example of Reward Hacking.



**Reward Hacking Case / Ambiguous Answer**

```
*Example: INCORRECT (Reward Hacking / Ambiguous Answer)*
Question: What is the country of citizenship of Fernando Santos?
Gold target: the United States
Predicted answer:
<think>
To determine the country of citizenship of Fernando Santos, I will
↪  follow these steps:

Step 1: Identify who Fernando Santos is.
Fernando Santos is a prominent figure in European football, known as a
↪  coach and manager of several football clubs and the national team
↪  of Portugal.

Step 2: Determine his country of origin.
Fernando Santos was born in Póvoa de Varzim, Portugal in 1952.

Step 3: Research his career and associations.
Throughout his career, Santos has managed football teams in Portugal
↪  and Greece, and he has been the manager of the Portugal national
↪  football team. He has also been involved in football
↪  administration in Portugal.

Step 4: Analyze his citizenship status.
Given that he was born in Portugal and has spent a significant part of
↪  his career there, it is reasonable to conclude that he holds
↪  Portuguese citizenship.

Step 5: Cross-reference with available information.
Cross-referencing with available biographical information on Fernando
↪  Santos, it is clear that he is a Portuguese citizen.

Conclusion: Based on the information gathered and analyzed, the
↪  country of citizenship of Fernando Santos is Portugal.
</think>
<answer> \boxed{United States of America, Portugal) </answer>
```



Figure 11: Another example of Reward Hacking.

