# OpenReview forum: "Edit-then-Consolidate for Reliable Knowledge Editing"
_ICLR.cc/2026/Conference — Submitted to ICLR 2026_

### Official Review · Reviewer_6WQv · 2025-10-16

**Soundness:** 3
**Presentation:** 3
**Contribution:** 3
**Rating:** 4
**Confidence:** 3

**Summary:**

This paper identifies a critical deficiency in most current methods within the LLM knowledge editing field: they focus solely on knowledge injection while neglecting the crucial step of knowledge consolidation. The authors experimentally demonstrate the importance of this consolidation phase, especially in lifelong editing scenarios. Consequently, the paper proposes a novel model editing paradigm, Edit-then-Consolidate, to bridge this gap. This framework employs a two-stage process involving Targeted Proximal Supervised Fine-Tuning for editing and GRPO for consolidation. Experiments confirm that the proposed method achieves strong performance in terms of reliability, generalization, and locality.

**Strengths:**

- I strongly agree with the paper's motivation. The poor performance of existing LLM knowledge editing methods under real-world evaluation settings is a significant and pressing issue in the field, and this paper makes a notable contribution toward addressing it.

- The addition of a second stage for knowledge consolidation following the initial edit is an elegant, practical, and highly intuitive approach.

- The experimental setup is rigorous and commendably uses an evaluation framework that more closely mirrors real-world application scenarios.

- The authors have provided a complete and detailed list of hyperparameter configurations, which is greatly appreciated for ensuring reproducibility.

**Weaknesses:**

- The paper lacks a discussion of the method's editing efficiency and time cost. A primary goal of knowledge editing is to update model knowledge with minimal computational resources and at a high speed. The use of GRPO for consolidation seems likely to introduce significant efficiency overhead, particularly when the number of editing samples is large. I believe it is necessary for the authors to include a discussion of this time-cost trade-off in the main text.

- The paper only presents results on a sequence of 1000 edits. However, in a true lifelong editing task, a model must handle a much larger volume of updates. Other works, such as AlphaEdit, RLEdit, and UltraEdit, have evaluated their methods on several thousands, or even tens of thousands of sequential edits. How does the Edit-then-Consolidate framework perform under these more demanding, larger-scale conditions?

- How are the specific FFN layers for TPSFT application chosen? Is this a manual selection process, or is there an automated method? How does the choice of different layers impact the method's overall performance?

- minor typos, e.g., "Fig. ref?" (line 184)

I promise to raise my rating if the authors address the concerns I have raised.

**Questions:**

See Weaknesses.

---

> ### Author Response · Authors · 2025-11-26
>
> > W1: *The paper lacks a discussion of the method's editing efficiency and time cost. A primary goal of knowledge editing is to update model knowledge with minimal computational resources and at a high speed. The use of GRPO for consolidation seems likely to introduce significant efficiency overhead, particularly when the number of editing samples is large. I believe it is necessary for the authors to include a discussion of this time-cost trade-off in the main text.*
>
> We thank the reviewer for highlighting the importance of efficiency. In the revision, we add an explicit **time–efficiency analysis** in **Appendix A.5 (Table 8 and Figure 9)** and reference it from the main text.
>
> **1. Effectiveness as a prerequisite for efficiency.** Efficiency is only meaningful if the edited knowledge remains usable. Under our fully autoregressive, LLM-as-judge evaluation, many existing “fast” editors exhibit **catastrophic failure** (e.g., MEMIT achieves nearly 0% Reliability on Qwen2.5-7B-Instruct, whereas EtCon reaches 69.4%; see Table.2). Thus, some extra computation is required to bridge the gap between “fast edits” and **reliably usable** knowledge in realistic settings.
>
>
> **2. Breakdown of time costs (Appendix A.5).** In our runtime study, we explicitly separate the **editing stage** and the **consolidation stage**.
> - **Editing stage.** We measure the average editing latency on Qwen2.5-7B-Instruct using the QAE dataset by performing 100 single-instance edits and recording the wall-clock time per edit for each method under the same configuration. As reported in **Table 8**, TPSFT attains an average editing time of **6.01 seconds per edit**, which is comparable to AlphaEdit (7.39 s) and MEMIT (7.78 s). GRACE (3.02 s) and WISE (2.68 s) are faster, while FT-M is the fastest (0.61 s). Overall, TPSFT’s per-edit cost is on par with representative parameter-editing baselines, indicating that its overhead in the editing stage remains moderate and compatible with practical deployment.
>
> | Methods              | **TPSFT (Ours)** | AlphaEdit | MEMIT | GRACE | WISE  | FT-M  |
> | -------------------- | ---------------- | --------- | ----- | ----- | ----- | ----- |
> | **Avg. Time / Edit** | **6.01s**        | 7.39s     | 7.78s | 3.02s | 2.68s | 0.61s |
>
> - **Consolidation stage.** Starting from edited LLM by FT-M, AlphaEdit, MMKE, and TPSFT, we then apply GRPO as a common consolidation algorithm on QAE using the **same hyperparameter configuration** for all methods and train for **15 optimization steps**, which corresponds to roughly **one hour of wall-clock time**. As shown in **Figure 9**, TPSFT+Con (EtCon) exhibits steadily increasing rewards and is close to convergence by step 15, whereas FT-M+Con and MMKE+Con improve more slowly and AlphaEdit+Con essentially fails to learn because the model has already collapsed during editing. Under identical GRPO training budgets, EtCon therefore achieves faster and more stable convergence in the consolidation stage.
>
> **3. The time–efficiency analysis in Appendix A.5** shows that, at the current experimental scale, the consolidation phase of EtCon incurs a lower time cost than several baselines while maintaining high editing reliability. The corresponding timing comparisons are reported in Appendix A.5 and explicitly referenced in the main text to clearly present EtCon’s time–cost trade-off.

---

> ### Author Response · Authors · 2025-11-26
>
> > W2: *The paper only presents results on a sequence of 1000 edits. However, in a true lifelong editing task, a model must handle a much larger volume of updates. Other works, such as AlphaEdit, RLEdit, and UltraEdit, have evaluated their methods on several thousands, or even tens of thousands of sequential edits. How does the Edit-then-Consolidate framework perform under these more demanding, larger-scale conditions?*
>
> Thank you for pointing this out.
>
> **Experiment setting**
> In the revision, we extend our sequential-editing experiments beyond 1k edits and explicitly evaluate EtCon under longer horizons. Specifically, **we have added new results in Appendix A.3 (Figure 7)** on a stream of **3,000 single-sample edits on ZsRE**. At each checkpoint (600, 1200, 1800, 2400, and 3000 edits), we first run the GRPO-based consolidation phase and then evaluate the model under our fully autoregressive, LLM-as-judge protocol before proceeding with further edits.
>
> **Experimental result analysis**
> The new curves show that **EtCon exhibits graceful degradation rather than catastrophic drift**: starting from 600 edits, Reliability and Generalization remain high and decrease only moderately as the number of edits triples, while Locality stays within a narrow band with no signs of collapse. In contrast, the FT-M baseline rapidly deteriorates as more edits accumulate, with Reliability and Generalization approaching near-zero and Locality dropping sharply. Thus, EtCon continues to outperform the baseline even after 3k sequential edits, supporting its suitability for larger-scale lifelong editing.
>
> Finally, we note that prior works reporting 10k+ edits typically use much cheaper teacher-forcing, token-level evaluations, whereas our evaluation combines TPSFT based knowledge editing, GRPO based knowledge consolidation, and fully autoregressive generation evaluated by LLM-as-judge (GPT-4.1). Under this substantially more expensive setting, assessing performance along trajectories up to 3k edits already requires significant compute, so we leave 10k+ edit trajectories to future work, while the reported 3k-edit results still show a clear advantage of EtCon over the baseline.

---

> ### Author Response · Authors · 2025-11-26
>
> > W3: *How are the specific FFN layers for TPSFT application chosen? Is this a manual selection process, or is there an automated method? How does the choice of different layers impact the method's overall performance?*
>
>
> Thank you for raising this point. In the revision, we clarify our TPSFT layer selection strategy and support it with empirical ablations.
>
> **Layer selection strategy.**
> Appendix A.1 now specifies that EtCon updates only the FFN down-projection layers (`mlp.down_proj`) in five blocks (layers 7–11 for Llama-3-8B-Instruct and 5–9 for Qwen2.5-7B-Instruct), following prior knowledge editing work [1-4] that targets FFN value projections as the main carriers of factual associations.
>
> **Ablation Experiment**
> (1) We add a new ablation in **Appendix A.2 (Table 6 and Figure 6)** where we edit three different 5-layer ranges on Llama-3-8B-Instruct: layers 7–11, 12–16, and 17–21, under identical hyperparameters. We observe that editing earlier layers (7–11) provides the best trade-off between Reliability/Generalization and Locality (73.5/63.1/30.2), while deeper layers (17–21) slightly improve Reliability but noticeably hurt Locality (17.3) and are more prone to reward hacking, as indicated by the “high reward but low performance’’ behavior in Figure 6.
>
> (2) We further validate this trend on a reasoning-oriented architecture in **Appendix A.4 (Table 7 and Figure 8)** by conducting the same ablation on DeepSeek-R1-Distill-Qwen-7B, editing layers 5–9, 13–17, and 23–27. Again, shallow layers (5–9) yield the best balance (88.6% Reliability, 53.5% Generalization, 17.0% Locality), while editing only deep layers reduces locality and leads to less stable consolidation dynamics.
>
> **Conclusion**
>
> Thus, the edited layer ranges are selected manually as a hyperparameter, guided by prior works and the above ablations. The new experiments show that EtCon is most effective when applied to these relatively earlier FFN blocks; while deeper blocks can also be edited, they tend to introduce additional reward hacking and locality degradation.
>
> [1] Geva, M., Schuster, R., Berant, J., & Levy, O. (2021, November). Transformer feed-forward layers are key-value memories. In _Proceedings of the 2021 Conference on Empirical Methods in Natural Language Processing_ (pp. 5484-5495).
>
> [2] Meng, K., Bau, D., Andonian, A., & Belinkov, Y. (2022). Locating and editing factual associations in gpt. _Advances in neural information processing systems_, _35_, 17359-17372.
>
> [3] Zhang, N., Yao, Y., Tian, B., Wang, P., Deng, S., Wang, M., ... & Chen, H. (2024). A comprehensive study of knowledge editing for large language models. _arXiv preprint arXiv:2401.01286_.
>
> [4] Qi, S., Yang, B., Jiang, K., Wang, X., Li, J., Zhong, Y., ... & Zheng, Z. In-Context Editing: Learning Knowledge from Self-Induced Distributions. In _The Thirteenth International Conference on Learning Representations_.
>
> > W4: *minor typos, e.g., "Fig. ref?" (line 184)?*
>
> We thank the reviewer for pointing this out. The “ref?” placeholder was caused by a compilation error in the citation and has been corrected in the revised version.

---

### Official Review · Reviewer_47g7 · 2025-10-25

**Soundness:** 2
**Presentation:** 3
**Contribution:** 2
**Rating:** 4
**Confidence:** 4

**Summary:**

The authors investigate why existing knowledge editing methods fail to maintain performance under continuous or lifelong knowledge updates. They focus on approaches that directly modify model parameters and demonstrate that the issue stems not from data insufficiency but from structural limitations, overfitting to new facts and the absence of a knowledge consolidation phase. As a result, models end up “storing” information without truly integrating it into their reasoning behavior.

To address this, the authors propose the Edit-then-Consolidate (EtCon) framework, introducing a consolidation stage that combines Targeted Proximal Supervised Fine-Tuning (TPSFT) and Group Relative Policy Optimization (GRPO) to bridge the gap between parametric knowledge and reasoning policy. Empirical results on Llama-3 and Qwen-2.5 show that EtCon substantially improves both reliability and generalization compared with prior editing methods.

**Strengths:**

1. The paper correctly identifies the limitations of existing knowledge editing methods, which tend to perform only fragmentary updates without ensuring integration into the model’s reasoning process. Its methodological setup, which assumes more realistic, real-world editing scenarios, is both coherent and practically motivated.

2. By combining TPSFT and GRPO, the authors propose an interesting solution that directly addresses a critical gap — the failure of parametric updates to propagate into the model’s reasoning policy.

3. The method achieves substantial performance gains under sequential editing conditions, indicating its robustness in lifelong or continual update settings.

4. Overall, the paper is well-structured, and the inclusion of ablation experiments and analyses of reward hacking effectively reinforce the authors’ main claims and lend additional credibility to the proposed framework.

**Weaknesses:**

1. According to the taxonomy presented by the authors, the comparison with memory-based or meta-learning editing methods is incomplete. For example, systems such as GRACE [1] and RECIPE [2] should have been included to provide a fairer and more comprehensive empirical evaluation.

[1] Hartvigsen, T., Sankaranarayanan, S., Palangi, H., Kim, Y., & Ghassemi, M. (2023). Aging with grace: Lifelong model editing with discrete key-value adaptors. Advances in Neural Information Processing Systems, 36, 47934-47959.

[2] Chen, Q., Zhang, T., He, X., Li, D., Wang, C., & Huang, L. (2024, November). Lifelong Knowledge Editing for LLMs with Retrieval-Augmented Continuous Prompt Learning. In Proceedings of the 2024 Conference on Empirical Methods in Natural Language Processing (pp. 13565-13580).

2. The notion of a trust region in TPSFT remains ambiguous. The paper does not provide any mathematical justification or upper-bound analysis explaining how the choice of ε ensures locality or under what conditions the gradient drift can be effectively controlled. Consequently, the claim that clipping “empirically works” is not theoretically substantiated.

3. The rationale for layer selection in TPSFT is also unclear. The choice appears to rely on the general assumption that lower layers encode syntactic information while upper layers capture semantics, yet there is no empirical evidence such as layer-wise activation or factual neuron analysis supporting this decision.

4. Although the authors present experiments under the heading “Sequential Editing,” the experimental configuration lacks sufficient detail. It is unclear how many edits are applied per model instance, whether updates occur in batch or single-shot form, and how the evaluation accounts for interaction effects among edits.

5. The paper does not validate whether referencing only the immediately preceding model state for policy updates (policy-update chaining) is the most appropriate strategy. It also omits discussion of how the method handles interference or conflict with much earlier edits, which is critical for lifelong learning stability.

6. Finally, the logic behind the GRPO reward design is under-explained. The paper lists accuracy, format, cleanliness, and consistency as reward components but provides little theoretical reasoning for their combination, weighting, or potential trade-offs.

**Questions:**

1. A placeholder “ref?” appears at line 184 and should be corrected.

2. Please clarify whether there are potential conflicts or trade-offs among the GRPO reward components.

3. Provide deeper analysis of the CoT-based training strategy. How does the use of chain-of-thought reasoning affect performance in reasoning-centric or multi-step models? Would the results differ when applied to inherently reasoning-oriented architectures?

---

> ### Author Response · Authors · 2025-11-26
>
> > W1: *According to the taxonomy presented by the authors, the comparison with memory-based or meta-learning editing methods is incomplete. For example, systems such as GRACE [1] and RECIPE [2] should have been included to provide a fairer and more comprehensive empirical evaluation.*
>
> We thank the reviewer for the valuable suggestion regarding the inclusion of memory-based and meta-learning baselines.
>
> **Experimental detail**
>
> To ensure a comprehensive evaluation, we have conducted additional experiments with **GRACE** [1] and **RECIPE** [2] on Qwen2.5-7B-Instruct under our lifelong sequential editing setup of **1,000 sequential single-sample edits**, across ZsRE, COUNTERFACT, and QAEdit.
>
>
> **Experimental results and analysis.**
>
> As shown in table below, the two methods exhibit distinct trade-offs:
> - **GRACE:** While it achieves higher **Reliability** scores than EtCon and maintains competitive **Locality**, its **Generalization** capability is severely limited (e.g., dropping to single digits). This suggests a significant gap in practical utility where generalization is required.
> - **RECIPE:** This method demonstrates good **Locality**, but its **Reliability** and **Generalization** are extremely limited.
> These additional comparisons further validate the balanced performance of our proposed EtCon method.
>
> | **Method**    | **ZsRE (Reli.)** | **ZsRE (Gen.)** | **ZsRE (Loc.)** | **COUNTERFACT (Reli.)** | **COUNTERFACT (Gen.)** | **COUNTERFACT (Loc.)** | **QAEdit (Reli.)** | **QAEdit (Gen.)** | **QAEdit (Loc.)** |
> | ------------- | ---------------- | --------------- | --------------- | ----------------------- | ---------------------- | ---------------------- | ------------------ | ----------------- | ----------------- |
> | **Pre-edit**  | 4.4              | 3.2             | **28.5**        | 1.0                     | 0.5                    | **36.9**               | 9.8                | 10.1              | **36.2**          |
> | **FT-M**      | 5.6              | 5.5             | 23.1            | 3.2                     | 3.1             | 24.4                   | 14.6               | 14.5      | 30.7              |
> | **MEMIT**     | 0.0              | 0.1             | 0.0             | 0.0                     | 0.2                    | 0.1                    | 0.4                | 0.3               | 0.2               |
> | **ALPHAEDIT** | 15.9             | 11.5     | 6.8             | 0.0                     | 0.0                    | 0.0                    | 0.0                | 0.0               | 0.0               |
> | **WISE**      | 4.5              | 3.3             | 19.1            | 1.4                     | 1.5                    | 31.0            | 7.1                | 9.7               | 16.9              |
> | **GRACE**     | **77.9**         | 3.1             | 20.0            | **82.9**                | 0.5                    | 26.8                   | **81.9**           | 8.7               | 16.1              |
> | **RECIPE**    | 4.0              | 3.5             | 23.7            | 1.7                     | 1.2                    | 17.8                   | 8.3                | 8.2               | 24.1              |
> | **EtCon**     | 69.4     | **60.8**        | 24.4     | 59.6             | **43.2**               | 29.7                   | 75.1        | **63.0**          | 32.3       |

---

> ### Author Response · Authors · 2025-11-26
>
> > W2: *The notion of a trust region in TPSFT remains ambiguous. The paper does not provide any mathematical justification or upper-bound analysis explaining how the choice of ε ensures locality or under what conditions the gradient drift can be effectively controlled. Consequently, the claim that clipping “empirically works” is not theoretically substantiated.*
>
> We thank the reviewer for raising this point. We clarify the notion of a “trust region” in TPSFT and provide both a simple gradient bound and empirical evidence.
>
>
> 1. **Policy-space trust region via ratio clipping.**
>    TPSFT optimizes the objective over the FFN weights $\theta_{\mathrm{FFN}}$:
>
> L_TPSFT(θ_FFN) = - 𝔼_((S_t,a_t)~D) [ min ( r_t(θ_FFN), clip(r_t(θ_FFN), 1-ε, 1+ε) ) ]
>
>    where $r_t(\theta_{\mathrm{FFN}}) = \frac{\pi_{\theta_{\mathrm{FFN}}}(a_t \mid S_t)}{\pi_{\mathrm{ref}}(a_t \mid S_t)}$.
>    In this formulation, the “trust region” is defined in **policy space** by constraining the probability ratio around 1.
>
> 2. **Simple bound on gradient drift.**
>    For all tokens that contribute non-zero gradient, the surrogate enforces $0 \le r_t(\theta_{\mathrm{FFN}}) \le 1+\epsilon$ (when $r_t > 1+\epsilon$ the objective saturates and the gradient from that trajectory becomes zero). Hence the per-sample TPSFT gradient can be written as
>
>    $$
>    \nabla_{\theta_{\mathrm{FFN}}} L_{\text{TPSFT}}
>    = - \mathbb{E}[r_t(\theta_{\mathrm{FFN}}) \nabla_{\theta_{\mathrm{FFN}}} \log \pi_{\theta_{\mathrm{FFN}}}(a_t \mid S_t)],
>    $$
>
>    implying the norm is bounded as
>
>    $$
>    \|\nabla_{\theta_{\mathrm{FFN}}} L_{\text{TPSFT}}\|
>    \le (1+\epsilon) \mathbb{E}\big[\|\nabla_{\theta_{\mathrm{FFN}}} \log \pi_{\theta_{\mathrm{FFN}}}(a_t \mid S_t)\|\big].
>    $$
>
>    This implies that each TPSFT step can only induce a bounded change in the log-probabilities, which constrains per-edit policy drift and thus serves as a local trust-region–style control, while trajectories with $r_t > 1+\epsilon$  are effectively ignored.
>
>
> 3. **Empirical evidence that clipping controls locality.**
>    Beyond this mild analysis, our main claim about the effectiveness of clipping is empirical. In Table 4, we directly compare editing with standard SFT versus TPSFT under the same architecture and data. While both achieve similarly low editing success in isolation, TPSFT significantly better preserves general capabilities (e.g., on downstream QA and reasoning benchmarks), whereas SFT causes substantial degradation.
> **In the revision, Appendix A.6 (Table 9) additionally reports an ablation over the $\epsilon$**,
> showing that our chosen provide the best trade-off between Reliability, Generalization, and Locality,
> which is consistent with our trust-region interpretation of TPSFT.
> Taken together with the gradient bound above, this provides empirical evidence that the ratio-based objective helps limit destructive policy drift during editing.

---

> ### Author Response · Authors · 2025-11-26
>
> > W3: *The rationale for layer selection in TPSFT is also unclear. The choice appears to rely on the general assumption that lower layers encode syntactic information while upper layers capture semantics, yet there is no empirical evidence such as layer-wise activation or factual neuron analysis supporting this decision.*
>
>
> Thank you for raising this point. In the revision, we clarify our TPSFT layer selection strategy and support it with empirical ablations.
>
> **Layer selection strategy.**
> In this work, we **follow prior knowledge editing work**. As clarified in the revision (Appendix A.1), EtCon updates only the FFN down-projection layers (`mlp.down_proj`) in layers 7–11 for Llama-3-8B-Instruct and 5–9 for Qwen2.5-7B-Instruct.
>
> **New layer ablations in the revision.**
> To provide empirical support for this choice, we additionally add **layer ablations** in the appendix. **We have conducted new ablation studies in the newly added Appendix A.2**, where we fix all hyperparameters on Llama-3-8B-Instruct and compare editing three 5-layer ranges (7–11, 12–16, 17–21). In **Appendix A.4**, we run the same type of ablation on DeepSeek-R1-Distill-Qwen-7B (5–9, 13–17, 23–27). In both models, the ranges used in our main experiments (7–11 and 5–9) yield the best trade-off between Reliability, Generalization, and Locality, while shifting edits to deeper layers consistently harms locality and stability.
>
> **Future work.**
> Therefore, the layer choice is both followed by prior work and empirically supported by our layer ablations. Our core goal in this paper is to enable **reliable knowledge editing in realistic autoregressive settings** via a two-stage Edit-then-Consolidate framework. A more fine-grained, neuron-level analysis of layer-wise “factual units” is important but orthogonal to this objective, and we view it as valuable future work built on top of our framework.
>
>
> > W4: *Although the authors present experiments under the heading “Sequential Editing,” the experimental configuration lacks sufficient detail. It is unclear how many edits are applied per model instance, whether updates occur in batch or single-shot form, and how the evaluation accounts for interaction effects among edits.*
>
> Thank you for pointing out the missing details in our “Sequential Editing” configuration. We clarify our setup as follows.
>
> **Sequential editing setup.**
> For each base model (Llama-3-8B-Instruct and Qwen-2.5-7B-Instruct) and each benchmark (ZsRE, COUNTERFACT, QAEdit), we perform **1000 sequential edits on a single model instance**. The edits are applied **one fact at a time** with **edit batch size = 1** during TPSFT, so each new edit starts from the parameters produced by the previous one.
>
> **Accounting for interaction effects among edits.**
> In our main editing results (Table 2 and Table 3), the editing metrics (Reliability, Generalization, Locality) are computed **after this full sequence of 1,000 sequential edits followed by the GRPO-based consolidation stage** on the same model instance, so they already reflect the cumulative interaction among edits. To further probe long-horizon effects, Appendix A.3 (Figure 7) extends the ZsRE setup to **3,000 edits** and evaluates performance at intermediate checkpoints (600/1200/1800/2400/3000), showing that **EtCon remains stable and consistently outperforms the FT-M baseline even under thousands of sequential edits**.

---

> ### Author Response · Authors · 2025-11-26
>
> > W5: *The paper does not validate whether referencing only the immediately preceding model state for policy updates (policy-update chaining) is the most appropriate strategy. It also omits discussion of how the method handles interference or conflict with much earlier edits, which is critical for lifelong learning stability.*
>
> We thank the reviewer for this comment and clarify our design choices.
>
> **Policy-update chaining strategy.**
> Our choice to reference only the immediately preceding model in TPSFT/GRPO is a deliberate design for locality and compositionality: at edit step $k$ we clip the likelihood ratio $r_t = \pi_{\theta^{(k)}}(a_t \mid S_t) / \pi_{\theta^{(k-1)}}(a_t \mid S_t)$ around 1, so each edit is a small, controlled update on top of all previous edits. Using a fixed reference (e.g., the initial model) across hundreds of edits would make $r_t$ compound, forcing either vanishingly small updates or a very large $\epsilon$, which we found to be less stable in preliminary experiments. **In the revision, Appendix A.6 (Table 10) additionally presents an ablation over the reference-model update interval**, showing that our policy-update chaining configuration achieves higher Reliability and Generalization than less frequent updates, providing further empirical support for this design in our setting.
>  We do not claim policy-update chaining is the unique optimal strategy, and we will clarify this as a practical design choice and leave systematic comparison to future work.
>
> **Controlling interference with earlier edits.**
> With ratio clipping, only trajectories with $0 \le r_t \le 1+\epsilon$ contribute non-zero gradients; for these, the per-edit change in log-probability is bounded by $|\log \pi_{\theta^{(k)}}(a_t \mid S_t) - \log \pi_{\theta^{(k-1)}}(a_t \mid S_t)| \le \log(1+\epsilon)$, while trajectories with excessively large ratios are ignored. Thus, each edit can only make a bounded change to the trajectory probability, helping to prevent large sudden shifts even over many steps.  The consolidation stage is then trained on queries that jointly cover all edited facts, which further stabilizes the policy and reduces interference between earlier and later edits.
>
> **Empirical evidence for long-horizon stability.**
> In our lifelong experiments with 1,000 sequential edits (Tables 2 and 3), EtCon maintains substantially higher Reliability and better Locality than strong editing baselines. **Appendix A.3 (Figure 7) additionally reports experiments with 3,000 sequential edits** and shows that EtCon remains stable and achieves better performance than FT-M in our experiments. We have made this empirical and analytical justification explicit in the revision.
>
>
> > W6: *Finally, the logic behind the GRPO reward design is under-explained. The paper lists accuracy, format, cleanliness, and consistency as reward components but provides little theoretical reasoning for their combination, weighting, or potential trade-offs.*
>
> > Q2：Please clarify whether there are potential conflicts or trade-offs among the GRPO reward components.
>
>
> Thank you for this insightful comment.
>
> **Empirical motivation.**
> We initially used only accuracy and format as reward components, but empirically observed reward hacking under this design. We therefore added cleanliness and consistency as auxiliary terms to penalize degenerate yet high-scoring outputs (e.g., noisy or contradictory generations) and reduce such reward hacking in practice. We do not claim this particular combination or weighting to be theoretically optimal.
>
> **Future work.**
> The main goal of this work is to propose the *Edit-then-Consolidate* paradigm for realistic knowledge editing, rather than to optimize a specific benchmark metric via a bespoke reward. GRPO is one practical instantiation of our consolidation stage, and exploring alternative reward decompositions, weightings, and theoretical trade-offs is an interesting direction for future work.
>
> > Q1: *"A placeholder “ref?” appears at line 184 and should be corrected.".*
>
> We thank the reviewer for pointing this out. The “ref?” placeholder was caused by a compilation error in the citation and has been corrected in the revised version.

---

> ### Author Response · Authors · 2025-11-26
>
> > Q3: Provide deeper analysis of the CoT-based training strategy. How does the use of chain-of-thought reasoning affect performance in reasoning-centric or multi-step models? Would the results differ when applied to inherently reasoning-oriented architectures?
>
> Thank you for this thoughtful question. In the revision, we clarify the role of CoT in TPSFT and add experiments on a reasoning-oriented architecture to directly address your concerns (Appendix A.4 and A.7).
>
> **CoT-based training strategy and motivation.**
> Our goal is to use smoother, self-induced targets to stabilize knowledge edits, rather than enforcing an extremely sharp one-hot constraint. Prior work on knowledge editing shows that directly fitting one-hot supervision on the edited answer can cause severe overfitting or even model collapse, whereas imitating a self-induced distribution yields more stable behavior [1–3]. Following this line, Appendix A.7 details how we construct CoT-based targets: for each editing instance, we build a **guiding context** $c$ that explicitly encodes the updated fact, query a frozen vanilla LLM with $[c, S_i]$ to obtain a step-by-step CoT, and **keep the CoT but overwrite the final answer span with the ground-truth {new\_target}**, discarding and regenerating CoTs that are clearly inconsistent with the injected fact. TPSFT then minimizes the discrepancy between the edited model distribution $\pi_{\theta_{\text{new}}}(\cdot \mid S_i)$ and this self-induced teacher trajectory $y_i^\star$ **under a ratio-clipping constraint**, so that CoT shapes a smoother reasoning trajectory while the answer supervision remains strictly correct, which empirically leads to more stable and reliable edits.
>
> **Results on reasoning-oriented architectures.**
> To study the effect of this CoT-based strategy on inherently reasoning-centric models, we extend our evaluation to **DeepSeek-R1-Distill-Qwen-7B** in **Appendix A.4**, using 1,000 ZsRE edits. As reported in Table 7 and Figure 8, EtCon achieves **88.6% Reliability, 53.5% Generalization, and 17.0% Locality** when editing layers 5–9, exhibiting a trade-off pattern similar to that on standard instruction-tuned LLMs. Training curves remain stable, whereas moving edits to deeper layers yields less favorable locality and convergence. Overall, we do not observe degradation of multi-step reasoning; instead, the CoT-based TPSFT + consolidation pipeline transfers well to a reasoning-oriented architecture and continues to integrate edited knowledge into its reasoning behavior.
>
> [1] Scialanga et al., *SAKE: Steering Activations for Knowledge Editing*, ACL 2025.
>
> [2] Qi et al., *In-Context Editing: Learning Knowledge from Self-Induced Distributions*, ICLR 2025.
>
> [3] Liu, T., Li, R., Dong, Z., Liu, H., Tang, X., Yin, Q., ... & Gao, J. (2025). Mitigating Heterogeneous Token Overfitting in LLM Knowledge Editing. _Proceedings of Machine Learning Research_.

---

### Official Review · Reviewer_j3EJ · 2025-10-29

**Soundness:** 2
**Presentation:** 3
**Contribution:** 2
**Rating:** 4
**Confidence:** 4

**Summary:**

This paper attributes the significant performance gap between controlled teacher-forcing evaluations and realistic, auto-regressive, lifelong editing scenarios to two main factors: (1) edited models overfitting to new facts, thereby degrading pre-trained capabilities, and (2) the lack of a knowledge consolidation stage, which prevents new facts from integrating into the LLMs' reasoning policy. To address these issues, the authors propose Edit-then-Consolidate, a novel knowledge editing paradigm that mitigates overfitting through Targeted Proximal Supervised Fine-Tuning (TPSFT) and consolidates edited knowledge using Group Relative Policy Optimization (GRPO). Experimental results show that their framework improves editing performance in real-world evaluation settings.

**Strengths:**

1. This paper identifies an interesting and underexplored bottleneck in knowledge editing: the disconnect between parametric updates and reasoning behavior. The insight that successful editing requires not only injecting knowledge but also consolidating it into the model's reasoning policy is both novel and important.
2. This paper effectively leverages existing techniques from other fields (e.g., PSFT and GRPO) to address the challenges in knowledge editing, achieving notable performance gains.
3. The experimental settings for evaluating editing performance are comprehensive and well-defined, covering mainstream datasets, LLMs, and evaluation metrics.

**Weaknesses:**

1. This paper attributes the ineffectiveness of existing editing techniques to two factors: (1) edited models overfitting to new facts, and (2) the lack of a knowledge consolidation stage, but provides insufficient demonstration of these claims. Although the proposed EtCon framework enhances editing performance, this does not conclusively demonstrate that previous editing techniques necessarily lead to overfitting or that consolidation is a required process.
2. The implementation details of TPSFT are unclear and potentially risky. For example, while the paper states that TPSFT targets FFN parameters for editing, it does not specify which layers or whether all FFNs are updated. Moreover, directly replacing the generated answer with the new target fact in their original CoT data may be problematic, as such naive substitution could introduce contradictions or spurious influences that distort the model's reasoning patterns.
3. The authors employ CoT reasoning to answer the original atomic facts used for editing. It is unclear whether introducing CoT reasoning on such simple factual edits is necessary, as it may not reflect the genuine reasoning integration of the newly edited knowledge. A more straightforward way to evaluate whether the updated knowledge has been successfully integrated into the model's reasoning policy is to test performance on multi-hop questions related to the edited facts.

**Questions:**

1. Inconsistent statements. In Line 045, the authors state that "Knowledge editing methods can be categorized into **three** main paradigms," whereas in Line 118, they claim that "Knowledge editing methods for LLMs fall into **two** paradigms."
2. Typos. Line 183 "Fig. ref? ".
3. Why is the general capability performance (e.g., on benchmarks such as C-Eval) for the Consolidate stage missing in Table 4?

---

> ### Author Response · Authors · 2025-11-26
>
> > W1: *"1. This paper attributes the ineffectiveness of existing editing techniques to two factors: (1) edited models overfitting to new facts, and (2) the lack of a knowledge consolidation stage, but provides insufficient demonstration of these claims. Although the proposed EtCon framework enhances editing performance, this does not conclusively demonstrate that previous editing techniques necessarily lead to overfitting or that consolidation is a required process."*
>
> Thank you for this comment. We agree that our original wording was too absolute, and we have refined these claims in the revision.
>
> **1. On "edited models overfitting to new facts."**
> In the revision, we clarify that most of traditional methods _tend to_ overfit to the new fact, thereby degrading pre-trained capabilities.
>
> **2. On "the lack of a knowledge consolidation stage":**
> We position consolidation as an **empirically essential** stage for applying edited knowledge in fully autoregressive settings, rather than as a theoretical requirement. Our experiments show that parametric editing alone is often insufficient for autoregressive generation setting:
> - **Activating latent knowledge (Tables 1 and 3).**
>   As shown in Tables 1 and 3, baselines such as FT-M and AlphaEdit have low Reliability (e.g., 16.6% and 18.7% on Llama-3 in Table 1). Simply adding our consolidation stage on top of the same parametric edits raises Reliability to 62.9% and 50.4%, respectively. This suggests that the edits already encode the new facts, but a consolidation phase is needed to *activate* this latent knowledge for autoregressive generation setting.
> - **Dependency on parametric edits (Figure 2).**
>   Figure 2 shows that applying the consolidation algorithm directly to an unedited model yields only minor gains. This indicates that consolidation by itself cannot replace parametric editing and instead acts as a bridge that aligns the edited parameters with the inference policy.
>
> **Revision change.**
> We have updated the revision to state:
> > "This work's empirical analysis reveals two recurring issues associated with this gap: (1) Most of traditional methods lead the edited model to overfit to the new fact, thereby degrading pre-trained capabilities; (2) There is a critical absence of a knowledge consolidation stage, leaving new facts insufficiently integrated into LLMs' inference-time behavior under autoregressive generation, thereby leading to a mismatch between parametric knowledge and actual generation behavior.“

---

> ### Author Response · Authors · 2025-11-26
>
> > W2: *"2. The implementation details of TPSFT are unclear and potentially risky. For example, while the paper states that TPSFT targets FFN parameters for editing, it does not specify which layers or whether all FFNs are updated. Moreover, directly replacing the generated answer with the new target fact in their original CoT data may be problematic, as such naive substitution could introduce contradictions or spurious influences that distort the model's reasoning patterns.*
>
> Thank you for the comment. We have clarified both the edited parameter subset and the CoT construction in the revision.
>
> **Edited layers.**
> As detailed in Appendix A.1, TPSFT only updates the **down-projection layers** of the FFN blocks, and only within a narrow band of layers: 7–11 for Llama-3-8B-Instruct and 5–9 for Qwen2.5-7B-Instruct.
>
> **CoT construction and safeguards.**
> Our goal is to use smoother, self-induced targets to stabilize knowledge edits, rather than applying an extremely sharp one-hot constraint. Specifically, many fine-tuning–based methods minimize the discrepancy between the edited model’s prediction and a one-hot target distribution, which severe overfitting and even model collapse [1-3]. In contrast, we first construct an explicit context  that encodes the updated fact, query the frozen vanilla LLM with $[c, S_i]$ to generate a CoT, and then replace only its final answer span with the strictly correct new fact, obtaining a self-induced reasoning trajectory $y_i^\star$. **In practice, whenever a generated CoT is obviously inconsistent or incorrect with respect to the injected new fact (e.g., contradicting the target answer), we simply discard it and re-generate a new trajectory $y_i^\star$.** During TPSFT, we minimize the discrepancy between the edited model distribution $\pi_{\theta_{\text{new}}}(\cdot \mid S_i)$ and the self-induced teacher trajectory defined by  $y_i^\star$, while controlling the update magnitude via the PSFT ratio-clipping constraint. In this way, the parameter updates are guided to imitate the model’s full reasoning process under a context that contains the new fact, rather than being forcibly pushed toward a one-hot label on a single answer token, which empirically leads to more stable and reliable editing behavior in our experiments.
>
> [1] Scialanga, M., Laugel, T., Grari, V., & Detyniecki, M. (2025, July). SAKE: Steering Activations for Knowledge Editing. In Proceedings of the 63rd Annual Meeting of the Association for Computational Linguistics (pp. 15966-15978).
>
> [2] Qi, S., Yang, B., Jiang, K., Wang, X., Li, J., Zhong, Y., ... & Zheng, Z. In-Context Editing: Learning Knowledge from Self-Induced Distributions. In _The Thirteenth International Conference on Learning Representations_.
>
> [3] Liu, T., Li, R., Dong, Z., Liu, H., Tang, X., Yin, Q., ... & Gao, J. (2025). Mitigating Heterogeneous Token Overfitting in LLM Knowledge Editing. _Proceedings of Machine Learning Research_.

---

> ### Author Response · Authors · 2025-11-26
>
> > W3: *"The authors employ CoT reasoning to answer the original atomic facts used for editing. It is unclear whether introducing CoT reasoning on such simple factual edits is necessary, as it may not reflect the genuine reasoning integration of the newly edited knowledge. A more straightforward way to evaluate whether the updated knowledge has been successfully integrated into the model's reasoning policy is to test performance on multi-hop questions related to the edited facts.*
>
> Thank you for this insightful comment. We clarify the role of CoT and add a portability evaluation.
>
>
> **Role of CoT in TPSFT.**
> Our goal is not to argue that simple atomic facts intrinsically require CoT. In the edit stage, we let the *vanilla* model generate a natural reasoning trace for each new fact, yielding a self-induced trajectory $y_i^\star$. These CoT-augmented labels smooth the supervision signal and preserve the model’s native reasoning style, instead of enforcing a sharp one-hot target on a few answer tokens. Prior work has shown that such smoothed, self-induced supervision can improve stability in knowledge editing [1,2]. Unlike those approaches, we do **not** introduce any external teacher model: all CoTs are produced by the same model being edited, avoiding conflicts between external teachers.
>
> **New portability evaluation.**
> As shown in below table, under our real-world evaluation setting, we further evaluate baselines on ZsRE with Llama-3-8B-Instruct as the base model. On multi-hop questions, EtCon still achieves a 7.4% advantage in Portability over the best baseline.
>
> | **Method** | **ZsRE (Reli.)** | **ZsRE (Gen.)** | **ZsRE (Loc.)** | **ZsRE (Port.)** |
> | ---------- | ---------------- | --------------- | --------------- | ---------------- |
> | FT-M       | 16.6             | 15.5            | 29.3            | 18.8             |
> | MEMIT      | 0.1              | 0.1             | 0.0             | 0.8              |
> | ALPHAEDIT  | 18.7             | 14.0            | 6.3             | 7.4              |
> | EtCon      | **73.5**         | **63.1**        | **30.2**        | **26.2**         |
> |            |                  |                 |                 |                  |
>
> [1] Scialanga, M., Laugel, T., Grari, V., & Detyniecki, M. (2025, July). SAKE: Steering Activations for Knowledge Editing. In Proceedings of the 63rd Annual Meeting of the Association for Computational Linguistics (pp. 15966-15978).
>
> [2] Qi, S., Yang, B., Jiang, K., Wang, X., Li, J., Zhong, Y., ... & Zheng, Z. In-Context Editing: Learning Knowledge from Self-Induced Distributions. In _The Thirteenth International Conference on Learning Representations_.
>
>
> > Q1: *"Inconsistent statements. In Line 045, the authors state that "Knowledge editing methods can be categorized into **three** main paradigms," whereas in Line 118, they claim that "Knowledge editing methods for LLMs fall into **two** paradigms.*
>
>
> Thank you for pointing this out. In the revision, we keep the three-paradigm categorization in Line 045 and rewrite the sentence at Line 118 to say that these three paradigms can be coarsely grouped into two architectural families (parametric in-place vs external-assisted), thereby removing the inconsistency.
>
>
>
> > Q2: *"Typos. Line 183 "Fig. ref? ".*
>
> We thank the reviewer for pointing this out. The “ref?” placeholder was caused by a compilation error in the citation and has been corrected in the revised version.
>
> > Q3: *"Why is the general capability performance (e.g., on benchmarks such as C-Eval) for the Consolidate stage missing in Table 4?".*
>
> Thank you for pointing this out.
>
> In the revised manuscript, we have added the general capability results (e.g., C-Eval) for the Consolidate stage to Table 4. As reported in Table 3, our experiments show that using GRPO as the knowledge consolidation algorithm almost never harms the model’s general capabilities and can even slightly improve them (excluding pathological cases where a specific editing method causes the edited model to collapse). The newly added results in Table 4 are consistent with this observation and provide a more complete comparison of performance before and after the Consolidate stage.

---

> > ### Comment · Reviewer_j3EJ · 2025-11-28
> >
> > Thank you for the comprehensive rebuttal. The clarifications on the TPSFT implementation and the additional portability results have addressed most of my concerns. I intended to raise my score to 6.
> >
> > However, due to the current system restrictions, I am unable to update the score field directly.
> > I am noting this here and hope the Area Chair takes it into account when making the final decision.
> >
> > Good luck with the paper!

---

> > > ### Author Response · Authors · 2025-12-01
> > >
> > > Thank you very much for your thoughtful comments and recognition of the improvements in our revisions. We sincerely appreciate your constructive suggestions, which have greatly enhanced this paper. Your positive feedback and support have greatly encouraged us. Thank you again for your time and contribution to our work, we will thank all the reviewers in our acknowledgements.

---

### Official Review · Reviewer_L4h6 · 2025-10-30

**Soundness:** 2
**Presentation:** 3
**Contribution:** 3
**Rating:** 4
**Confidence:** 4

**Summary:**

The paper argues that the gap between teacher-forcing knowledge–editing results and real-world, auto-regressive performance stems from missing consolidation: after injecting a fact into the weights, models often fail to use it consistently at inference time. The authors propose a two-stage pipeline: (i) Targeted Proximal SFT (TPSFT), which edits only selected FFN layers under a trust-region objective and with CoT-augmented labels; and (ii) a GRPO-based consolidation step with a composite reward (accuracy/format/cleanliness/consistency). Across ZsRE, CounterFact, and QAEdit on Llama-3-8B-Instruct and Qwen2.5-7B-Instruct, EtCon reports large gains in Reliability and Generalization while maintaining Locality and general capabilities

**Strengths:**

(1) In autoregressive generation, the paper shows that this missing consolidation stage is the main bottleneck for current methods and argues that editing should be a two-stage process: first injecting the new fact into model weights, then explicitly consolidating it so the model’s reasoning policy actually uses that fact.
(2) Under sequential auto-regressive evaluation across ZsRE, COUNTERFACT, and QAEdit, EtCon substantially outperforms baselines like FT-M, WISE, MEMIT, and ALPHAEDIT on Reliability and Generalization, while keeping acceptable Locality and avoiding catastrophic collapse of pretrained capabilities.

**Weaknesses:**

(1) The paper reports R/G/Locality and broad general-ability benchmarks, but lacks an explicit portability test showing that the newly edited fact transfers to broader downstream tasks which weakens the claim that consolidation improves real reasoning with the new fact.
(2) The paper states that existing methods “overfit to new facts,” degrading general capabilities, but the presented results often show something different: many baselines simply fail to perform the edit at all (e.g., FT-M and WISE have single-digit Reliability on Qwen2.5-7B). This looks less like classic overfitting-to-the-new-fact (high edit success but poor locality) and more like failure to edit or model collapse, so the causal narrative around “overfitting” should be clarified.
(3) The paper repeatedly frames the setting as “lifelong” and “sequential,” and it reports results over 1000 sampled instances per dataset, but it never clearly states how many edits are actually applied in sequence to the same model before evaluation. Without stating the length of the edit trajectory or showing performance as the number of edits grows, it’s hard to judge long-term stability.

**Questions:**

See Weaknesses.

---

> ### Author Response · Authors · 2025-11-26
>
> > W1: *"The paper reports R/G/Locality and broad general-ability benchmarks, but lacks an explicit portability test showing that the newly edited fact transfers to broader downstream tasks which weakens the claim that consolidation improves real reasoning with the new fact. "*
>
> Thank you for raising this insightful concern.
>
> **ZsRE portability evaluation.**
> We add an explicit portability evaluation on ZsRE, denoted **ZsRE (Port.)**, which measures whether the edited fact is used in multi-hop questions. We report results on **Llama-3-8B-Instruct** under the same editing and consolidation setup as in Table 2.
>
> **Experimental analysis.**
> As shown in the table below, EtCon reaches **26.2%** portability, outperforming FT-M (**18.8%**), ALPHAEDIT (**7.4%**), and MEMIT (**0.8%**). Together with its much higher Reliability and Generalization on ZsRE, this supports our claim that consolidation helps the model actually _use_ the edited fact in downstream reasoning tasks.
>
> | **Method** | **ZsRE (Reli.)** | **ZsRE (Gen.)** | **ZsRE (Loc.)** | **ZsRE (Port.)** |
> | ---------- | ---------------- | --------------- | --------------- | ---------------- |
> | FT-M       | 16.6             | 15.5     | 29.3            | 18.8      |
> | MEMIT      | 0.1              | 0.1             | 0.0             | 0.8              |
> | ALPHAEDIT  | 18.7      | 14.0            | 6.3             | 7.4              |
> | EtCon      | **73.5**         | **63.1**        | 30.2     | **26.2**         |
>
> **Future work.**
> The absolute portability gain is still modest compared to the improvement on single-hop Reliability. We believe this is because EtCon is trained only on single-hop QA derived from the edited facts, without explicit supervision on multi-hop compositions. Extending EtCon with such multi-hop supervision is a natural next step to further improve portability.
>
> > W2: *"The paper states that existing methods “overfit to new facts,” degrading general capabilities, but the presented results often show something different: many baselines simply fail to perform the edit at all (e.g., FT-M and WISE have single-digit Reliability on Qwen2.5-7B). This looks less like classic overfitting-to-the-new-fact (high edit success but poor locality) and more like failure to edit or model collapse, so the causal narrative around “overfitting” should be clarified.? "*
>
>
> Thank you for pointing this out; our wording around “overfitting to new facts” was imprecise, and we have revised it in the revision.
>
> **Clarifying our usage.**
> In our paper, *overfitting to new facts* does not mean the classic pattern of “high edit success but poor locality.” We use it to refer to parameter updates that over-specialize the model to the edited facts and prompts, leading to two phenomena:
> 1. **Parametric update misaligned with autoregressive generation behavior.**
>    The new fact is successfully written into the model parameters, but the model fails to use it under fully autoregressive generation due to the lack of consolidation. Evidence:
> 	(1) [1],[2] consistently show these methods achieve high success rates under teacher-forcing evaluation. (2) As shown in **Table 2** and **Table 3**, adding our consolidation stage significantly boosts the autoregressive generation performance of existing methods (showing that the knowledge was successfully edited), yet the damage to pre-trained capabilities cannot be fully recovered.
> 2. **Model collapse.**
>    In more severe cases, the ripple effects of this parameter over-specialization completely destroy pre-trained capabilities. Even the knowledge consolidation stage cannot rectify this collapse (e.g., MEMIT and ALPHAEDIT in Table 2).
>
> **Revision change.**
> We have clarified this narrative in the revision by updating the text to:
> > "This work's empirical analysis reveals two recurring issues associated with this gap: (1) Most of traditional methods lead the edited model to overfit to the new fact, thereby degrading pre-trained capabilities; (2) There is a critical absence of a knowledge consolidation stage, leaving new facts insufficiently integrated into LLMs' inference-time behavior under autoregressive generation, thereby leading to a mismatch between parametric knowledge and actual generation behavior.“
>
> [1] Yang, W., Sun, F., Tan, J., Ma, X., Cao, Q., Yin, D., ... & Cheng, X. (2025). The Mirage of Model Editing: Revisiting Evaluation in the Wild. *CoRR*.
> [2] Fang, J., Jiang, H., Wang, K., Ma, Y., Shi, J., Wang, X., ... & Chua, T. S. AlphaEdit: Null-Space Constrained Knowledge Editing for Language Models. In *ICLR 2025*.

---

> ### Author Response · Authors · 2025-11-26
>
> > W3: *"The paper repeatedly frames the setting as “lifelong” and “sequential,” and it reports results over 1000 sampled instances per dataset, but it never clearly states how many edits are actually applied in sequence to the same model before evaluation. Without stating the length of the edit trajectory or showing performance as the number of edits grows, it’s hard to judge long-term stability. "
>
> Thank you for the comment.
>
> - **Sequential editing setup.**
>   We adopt a sequential editing setup: we apply **1,000 single-sample edits** to a single model instance.
>
> - **Evaluation point.**
>   Unless otherwise specified, the editing metrics in Tables 2 and 3 are computed **after these 1,000 edits followed by the GRPO-based consolidation stage**.
>
> - **Long-horizon stability.**
>   To assess long-term stability as the number of edits grows, **we have added new experiments in Appendix A.3 (Figure 7) to extend** the ZsRE stream to **3,000 sequential edits** with intermediate checkpoints (600/1200/1800/2400/3000), showing that **EtCon remains stable and consistently outperforms the FT-M baseline** even under thousands of edits.

---

### Author Response · Authors · 2025-12-03
**Final Summary [1/2]**

**Dear PCs, SACs, ACs, and Reviewers,**

**Thank you very much for your valuable contributions to our work.** To assist the newly assigned AC and help reduce their workload, we provide below a summary of the key points from the reviews and the reviewer-author discussions.

### 1. **Summary of Contribution**

This paper identifies that the fragile performance of most LLM knowledge editing methods under real-world evaluation settings stems from the **absence of a knowledge consolidation stage**. To address this, we propose **EtCon**, a novel two-stage knowledge editing paradigm that achieves **35%-50%** performance gains over strong baselines across mainstream datasets, LLMs, and real-world evaluation settings.

### 2. **Reviewer Status**
Below, we summarize the current status:

| Reviewer | Strengths                                                                                                                                                                                                       | Weakness/Question                                                                                                                                                                                                                                                                                                                | Attitude                                                                                       |
| :------- | :------------------------------------------------------------------------------------------------------------------------------------------------------------------------------------------------------------------------- | :------------------------------------------------------------------------------------------------------------------------------------------------------------------------------------------------------------------------------------------------------------------------------------------------------------------------------- | :--------------------------------------------------------------------------------------------- |
| **j3EJ** | Insight is "**novel and important**"; praised the **comprehensive and well-defined** experimental settings.                                                                                                                | Wording of claims; questioned necessity of CoT ; lack of portability test; Unclear FFN layer selection rationale; presentation issues                                                                                                                                                                                            | **Explicitly confirmed intention to raise score to 6.**                                        |
| **6WQv** | Addresses a "**significant and pressing issue**"; Paradigm is "**elegant, practical, and highly intuitive**"; Commended the **rigorous setup** mirroring real-world scenarios and reproducibility.                         | Lack of time-cost discussion; need evaluation under larger volumes of sequential edits ; unclear FFN layer selection rationale; presentation issues ;                                                                                                                                                                            | **Explicitly pledged to raise rating** contingent on addressing concerns (which we have done). |
| **47g7** | "**Correctly identifies limitations**" of prior methods; Methodology is "**coherent and practically motivated**"; "**Interesting solution**" bridging parametric knowledge and reasoning; Praised **effective ablations**. | Lack of baselines; lack of ablations on ε and reference model updates; under-explained GRPO reward design and trade-offs; lifelong editing setting detail; unclear FFN layer selection rationale; limited analysis of CoT-based training on reasoning-oriented models; minor citation placeholder; |                                                                                                |
| **L4h6** | "**Substantially outperforms baselines**" ; "Closely mirrors real-world scenarios"                                                                                                                                         | Lack of portability test; terminology precision; unclear sequential lifelong-editing setup and long-horizon stability                                                                                                                                                                                                            |                                                                                                |

**We believe we have addressed all of the above concerns and made the necessary revisions.**

---

> ### Author Response · Authors · 2025-12-03
> **Final Summary [2/2]**
>
> ### 3. **Revision Summary**
> **All reviewers agree that our work tackles an underexplored problem in LLM knowledge editing with a novel, straightforward paradigm.** To address specific concerns, we have made the following revisions (highlighted in blue in the paper):
>
> (1) Refined the wording “Existing methods overfitting to new facts”. (Reviewers L4h6, j3EJ)
>
> (2) Added ablation experiments on ε and reference-model update intervals in EtCon. (Reviewer 47g7)
>
> (3) Added layer-selection ablation experiments on both Llama-3-8B-Instruct and DeepSeek-R1-Distill-Qwen-7B and specified the FFN down-projection layers updated by TPSFT. (Reviewers j3EJ, 47g7, 6WQv)
>
> (4) Added a detailed runtime and efficiency analysis for both the editing stage and the consolidation stage. (Reviewer 6WQv)
>
> (5) Clarified CoT construction and added experiments on the reasoning-oriented DeepSeek-R1-Distill-Qwen-7B. (Reviewers j3EJ, 47g7)
>
> (6) Added GRACE and RECIPE as baselines. (Reviewer 47g7)
>
> (7) Added experiments under a larger-scale sequential editing setting. (Reviewers L4h6, 6WQv)
>
> (8) Clarified the sequential editing setup: single-sample edits with batch size 1 on the same model instance. (Reviewers L4h6, 47g7, 6WQv)
>
> (9) Added a portability evaluation to test multi-hop use of edited facts. (Reviewers L4h6, j3EJ)
>
> (10) Completed Table 4 and fixed minor inconsistencies. (Reviewers j3EJ, 47g7)
>
> In addition, we briefly summarize below how the key concerns of each reviewer have been addressed in the revision and rebuttal.
>
> ---
> **Reviewer L4h6**
>
> We have addressed all of Reviewer L4h6’s concerns in the revision and rebuttal:
>
> - **Portability (W1).** Added an explicit portability evaluation.
>
> - **Wording of “Existing methods overfitting to new facts”.  (W2).** Refined the wording to more precisely.
>
> - **Lifelong / sequential setup & stability (W3).** Clarified the 1,000-step same-model sequential editing setting and added larger-scale (3k-edit) sequential experiments.
>
> ---
> **Reviewer j3EJ**
>
> We have fully addressed Reviewer j3EJ’s concerns regarding TPSFT details, CoT design, portability evaluation, and presentation in the revision and rebuttal, and the reviewer has explicitly acknowledged this and confirmed their intention to raise the score to 6.
>
> ---
> **Reviewer 47g7**
>
> We have thoroughly addressed Reviewer 47g7’s concerns regarding missing baselines (adding GRACE and RECIPE; **W1**), the TPSFT ratio-clipping “trust region” (with new ablations on ε; **W2**), FFN layer selection (with new layer ablations; **W3**), the lifelong sequential editing setup (clarifying 1,000 single-sample edits on the same model instance; **W4**), the policy-update strategy (with new ablations on the reference-model update interval; **W5**), CoT-based training on reasoning-oriented models (by adding experiments; **Q3**), and presentation issues (**Q1**) through targeted revisions and detailed rebuttal clarifications.
>
> For the GRPO reward design (**W6, Q2**), we explained that the four-component reward is an empirically motivated way to mitigate reward hacking observed with accuracy-only signals. Our main contribution is the two-stage knowledge editing paradigm that reliably improves existing methods, rather than hand-tuning a bespoke reward to maximize a particular benchmark; we instead view alternative consolidation algorithms and more refined reward designs as natural directions for future work within this framework.
>
> We have fully addressed all of Reviewer 47g7’s concerns with additional experiments and clarifications in the revision and rebuttal; since they were unable to update their score due to system constraints, we would be grateful if the AC could take these additions into account when assessing our work.
>
> ---
> **Reviewer 6WQv**
>
> - **Efficiency (W1).** Added a detailed runtime and efficiency analysis.
>
> - **Larger-scale sequential editing (W2).** Extended the sequential editing experiments to 3k edits in the revision, showing stable long-horizon performance.
>
> - **Layer selection (W3).** Clarified the choice of edited FFN down-projection layers and added supporting layer-ablation experiments.
>
> The reviewer explicitly stated that they would raise their rating once these issues were addressed, which we have done in the revision.
>
> ---
>
> We highlight that Reviewer **j3EJ** explicitly confirmed their intention to raise the score to **6** but was unable to update it due to system restrictions, and Reviewer **6WQv** explicitly pledged to raise their rating contingent on our addressing efficiency and long-term stability, which we have done in the revision. For the remaining two reviewers, we have also carefully addressed all of their concerns, as reflected in the rebuttal and the revision. We would be very grateful if the AC could take these positive reassessments and our additional experiments into account in the final decision, and we sincerely thank the PCs and ACs for their time and careful consideration of our work.
>
> Sincerely,
> Authors

---

### Meta-Review · Area_Chair_VHHu · 2026-01-07

**Summary:**

In this work, the authors propose a model that sequentially edits a knowledge base and consolidates these changes to maintain coherence and factual consistency. The reviewers raised concerns primarily about whether the improvements observed in controlled settings transfer to broader, more complex tasks. They also questioned whether the distinction between overfitting and collapse in the baseline models was adequately explained and whether the scale of "lifelong" editing claims was sufficient. Additionally, reviewers flagged concerns about the heuristic nature of the reward design and the practical overhead of the consolidation process. While the authors have made several improvements in the rebuttal,  the core issues around portability and scalability persist. Therefore, the paper is recommended for future revisions.

**Reviewer Concerns:**

- Reviewer 47g7 and Reviewer L4h6 raised concerns about narrative mismatch: the paper claims ‘overfitting to new facts,’ but results look more like edit failure/collapse for some baselines. The authors revised the language to clarify the issue between overfitting and collapse, acknowledging the framing issue.
- Reviewer 47g7 raised concerns about the unclear sequential-edit scale (only 1000 edits, not truly lifelong). The authors partially addressed the concern by extending the experiments to 3000 sequential edits and providing additional analysis.
- Reviewer L4h6 raised concerns about how much computational cost the consolidation process adds. This was addressed by including a time-cost analysis
- Reviewer 47g7 is concerned with reward design: how weights/trade-offs among multiple reward terms are handled. The authors added ablations that explore the effect of different reward components and provided more detailed analysis of the reward terms.
- Reviewer 47g7 raised concerns on No ‘portability’ test showing how edited facts transfer to broader, multi-hop tasks. The concern remains outstanding as the issue of generalization to more complex tasks remains unresolved.

**Reviewer Scores:**

Reviewer j3EJ identified his/her intention to raise the score to 6. However, the conclusion is made post the server issue. No other intention of score changing is clearly made.

---

### Decision · Program_Chairs · 2026-01-26

Reject